# Intercomparison of methods to estimate gross primary production based on $CO_2$ and COS flux measurements

Kukka-Maaria Kohonen[1], Roderick Dewar[1,2], Gianluca Tramontana[3,4], Aleksanteri Mauranen[1], Pasi Kolari[1], Linda M.J. Kooijmans[5], Dario Papale[6,7], Timo Vesala[1,8,9], and Ivan Mammarella[1]

[1]Institute for Atmospheric and Earth System Research/ Physics, Faculty of Science, University of Helsinki, Helsinki, Finland
[2]Plant Sciences Division, Research School of Biology, The Australian National University, Canberra, ACT 2601, Australia
[3]Image Processing Laboratory (IPL), Parc Científic Universitat de València, Universitat de València, Paterna, Spain
[4]Terrasystem s.r.l, Viterbo, Italy
[5]Meteorology and Air Quality, Wageningen University and Research, Wageningen, The Netherlands
[6]DIBAF, Department for Innovation in Biological, Agro-food and Forestry Systems, University of Tuscia, Viterbo, Italy
[7]IAFES, Euro-Mediterranean Center on Climate Change (CMCC), Viterbo, Italy
[8]Institute for Atmospheric and Earth System Research / Forest Sciences, University of Helsinki, Helsinki, Finland
[9]Yugra State University, 628012, Khanty-Mansiysk, Russia

**Correspondence:** Kukka-Maaria Kohonen (kukka-maaria.kohonen@helsinki.fi)

**Abstract.** Separating the components of ecosystem scale carbon exchange is crucial in order to develop better models and future predictions of the terrestrial carbon cycle. However, there are several uncertainties and unknowns related to current photosynthesis estimates. In this study, we evaluate four different methods for estimating photosynthesis at a boreal forest at the ecosystem scale, of which two are based on carbon dioxide ($CO_2$) flux measurements and two on carbonyl sulfide (COS)
flux measurements. The $CO_2$-based methods use traditional flux partitioning and artificial neural networks to separate the net $CO_2$ flux into respiration and photosynthesis. The COS-based methods make use of a unique five-year COS flux data set and involve two different approaches to determine the leaf-scale relative uptake ratio of COS and $CO_2$ (LRU), of which one ($LRU_{CAP}$) was developed in this study. $LRU_{CAP}$ was based on a previously-tested stomatal optimization theory (CAP), while $LRU_{PAR}$ was based on an empirical relation to measured radiation.

For the measurement period 2013–2017, the artificial neural networks method gave a GPP estimate very close to that of traditional flux partitioning at all time scales. On average, the COS-based methods gave higher GPP estimates than the $CO_2$-based estimates on daily (23 and 7 % higher, using $LRU_{PAR}$ and $LRU_{CAP}$, respectively) and monthly scales (20 and 3 % higher), as well as a higher cumulative sum over three months in all years (on average 25 and 3 % higher). $LRU_{CAP}$ was higher than LRU estimated from chamber measurements at high radiation, leading to underestimation of midday GPP relative
to other GPP methods. In general, however, use of $LRU_{CAP}$ gave closer agreement with $CO_2$-based estimates of GPP than use of $LRU_{PAR}$. When extended to other sites, $LRU_{CAP}$ may be more robust than $LRU_{PAR}$ because it is based on a physiological model whose parameters can be estimated from simple measurements or obtained from the literature. In contrast, the empirical radiation relation in $LRU_{PAR}$ may be more site-specific. However, this requires further testing at other measurement sites.

# 1 Introduction

Photosynthetic carbon uptake (or gross primary production, GPP) is a key component of the global carbon cycle, with the terrestrial ecosystems removing approximately 30 % of annual anthropogenic carbon dioxide ($CO_2$) emissions from the atmosphere (Luo et al., 2015; Friedlingstein et al., 2020). With the current climatic warming it has been suggested that both photosynthesis and respiration are increasing, due to the $CO_2$ fertilization effect and rising temperatures providing more favourable conditions not only for photosynthesis but also for respiration (Dusenge et al., 2019). However, it is not known at which rate these

two processes are changing and thus the extent to which they offset each other. In addition, their relative importance varies seasonally, with photosynthesis predicted to increase more than respiration in spring, leading to greater carbon uptake, while respiration is predicted to increase more than photosynthesis in autumn, leading to net carbon emission in northern terrestrial ecosystems (Piao et al., 2008). Methods to measure and study photosynthesis and respiration individually are thus crucial for future carbon cycle predictions.

Eddy covariance (EC) is widely used to measure the biosphere-atmosphere exchange of $CO_2$ at the ecosystem scale. However, EC only measures net ecosystem $CO_2$ flux (NEE), which includes contributions from both $CO_2$ uptake by photosynthesis (GPP) and ecosystem respiration (R). Traditionally, NEE partitioning into GPP and respiration uses the method of Reichstein et al. (2005), in which temperature response curves are fitted to nighttime $CO_2$ flux data (respiration). However, this method relies on nighttime EC flux measurements, which are uncertain and often filtered out due to low turbulence conditions and

possible advective gas transport (Aubinet, 2008). To address this problem, partitioning methods have been developed based on a combination of nighttime temperature responses of respiration (as in nighttime method) and daytime radiation responses of GPP (daytime method) (Lasslop et al., 2010; Kulmala et al., 2019). However, both the nighttime method and the daytime method assume that respiratory processes operate in the same way during the day and night, and have uncertainties due to assumptions of functional relationships (Tramontana et al., 2020). These assumptions lead to uncertainties in partitioning be-

cause different biomass compartments (soil organic matter, roots, stems, branches, foliage) could have different drivers and respiration responses even within the same ecosystem (Kolari et al., 2009; Keenan et al., 2019). Leaf respiration during the day may be inhibited by radiation, the so-called Kok effect (Kok, 1949; Wohlfahrt et al., 2005; Heskel et al., 2013; Yin et al., 2020), and Keenan et al. (2019) and Wehr et al. (2016) suggest that, as a result, global GPP based on the nighttime method has been overestimated. On the other hand, photorespiration, which is an oxidation process competing with carboxylation under

radiation, might offset inhibition by the Kok effect (Heskel et al., 2013).

One way to address these uncertainties in flux partitioning is to use machine learning methods, such as artificial neural networks to separate NEE into respiration and GPP (Tramontana et al., 2020). The advantage of this method is that it makes no *a priori* assumptions about responses to environmental drivers but determines these based only on data. In a pioneering study, Desai et al. (2008) attempted to use artificial neural network to emulate the nighttime partitioning method, but obtained

no significant improvements. More recently, Tramontana et al. (2020) proposed a new approach ($NN_{C-part}$) involving novel methods for implementing the network's structure and of inferring GPP and R signals from NEE. Both nighttime and daytime NEE are used for network training, so the dynamics of biophysical processes are accounted for in a comprehensive way.

Yet another approach to addressing uncertainties in GPP estimates is to use proxies for photosynthetic $CO_2$ uptake. One such proxy is carbonyl sulfide (COS), which is a sulfur compound with a tropospheric mixing ratio of approximately 500 ppt (Montzka et al., 2007). While the use of different $CO_2$-based partitioning methods is primarily aimed at more accurate GPP estimation, in contrast the use of COS as a proxy for GPP is aimed at a better process understanding of GPP. COS is mainly produced by oceans and anthropogenic sources (Kettle et al., 2002; Berry et al., 2013; Launois et al., 2015; Whelan et al., 2018) while vegetation is the largest sink (Sandoval-Soto et al., 2005; Blonquist et al., 2011). COS has been proposed as a proxy for GPP because it is taken up by plants through the same diffusive pathway as $CO_2$ and transported to the chloroplast surface. There it is destroyed by a hydrolysis reaction catalyzed by the enzyme carbonic anhydrase (CA, also located within the cytoplasm (Polishchuk, 2021)), while $CO_2$ continues its journey inside the chloroplast, where it is assimilated in the Calvin cycle (Wohlfahrt et al., 2012). It is assumed that COS is completely removed by hydrolysis so that there is no back-flux from the leaf to the atmosphere (Protoschill-Krebs et al., 1996). Estimates of GPP from COS flux measurements use the leaf relative uptake ratio (LRU), that is, the ratio of COS and $CO_2$ deposition rates at the leaf scale. While LRU has been treated either as a global or plant-specific constant (Asaf et al., 2013; Stimler et al., 2012), recent studies have shown that LRU is a function of solar radiation because $CO_2$ uptake is highly radiation dependent while COS uptake is not (Stimler et al., 2010; Yang et al., 2018; Kooijmans et al., 2019; Spielmann et al., 2019), and may also vary with vapour pressure deficit (Sun et al., 2018b; Kooijmans et al., 2019). In addition to uncertainties related to variation in LRU, COS-based GPP estimates are uncertain because ecosystem-scale COS flux measurements typically have a low signal-to-noise ratio and high random uncertainty at 30 min timescale, although this is reduced when fluxes are averaged over longer time periods (Kohonen et al., 2020).

In this study, we compare the annual, seasonal, daily and sub-daily variation of i) a traditional GPP estimate ($GPP_{NLR}$, NLR referring to non-linear regressions) based on a combination of daytime and nighttime methods, ii) a neural network GPP estimate based on NEE and $NN_{C-part}$ ($GPP_{ANN}$), iii) a GPP estimate based on COS flux measurements using the radiation-dependent LRU function from Kooijmans et al. (2019) ($GPP_{COS,PAR}$) and iv) a GPP estimate based on COS flux measurements using a previously-published stomatal optimization model (CAP) to calculate LRU ($GPP_{COS,CAP}$) in a boreal evergreen needle-leaf forest during years 2013–2017. Our aim is to study potential inconsistencies in diel or seasonal patterns of GPP that may arise from extrapolating nighttime temperature responses to daytime, and to discuss the limitations and uncertainties of all four methods. We also make recommendations for improving COS-based GPP estimates.

## 2 Materials and methods

### 2.1 Site description

Measurements were conducted at the Hyytiälä forest Station for Measuring Ecosystem Atmosphere Relations (SMEAR) II measurement site (61°51'N, 24°17'E), where the forest stand is already more than 50 years old (Hari and Kulmala, 2005). The stand is dominated by Scots Pine (*Pinus Sylvestris L.*) with some Norway spruce (*Picea abies L. Karst.*) and deciduous trees (e.g. *Betula sp., Populus tremula, Sorbus aucuparia*). The daytime flux footprint covers c. 50 ha area of the forest. The canopy

height increased from approximately 18 to 20 m during the measurement period (2013–2017) and the all-sided leaf area index (LAI) was c. 8 $m^2$ $m^{-2}$.

## 2.2 Measurements

### 2.2.1 Eddy covariance fluxes and environmental measurements

EC measurements were made on a 23 m high tower. The setup consisted of a Gill HS (Gill Instruments Ltd., England, UK)
sonic anemometer measuring horizontal and vertical wind velocities and sonic temperature, and a quantum cascade laser (QCL; Aerodyne Research Inc., Billerica, MA, USA) for measuring COS, $CO_2$ and $H_2O$ mixing ratios at 10 Hz frequency. The setup is described in more detail in Kohonen et al. (2020) and flux data are presented in Vesala et al. (2022). Flux processing was done using EddyUH software (Mammarella et al., 2016) following the methods presented by Kohonen et al. (2020). Fluxes were corrected for storage change and filtered according to friction velocity. Storage change fluxes of COS were calculated from the
COS profile measurements in 2015–2017 and from concentration measurements at one height in other years, as described in Kohonen et al. (2020); $CO_2$ storage change fluxes were calculated from $CO_2$ concentration profile measurements. The friction velocity threshold was determined from $CO_2$ fluxes (Papale et al., 2006) and a threshold of 0.3 m $s^{-1}$ was applied to the entire data set to exclude periods of low turbulence. COS flux processing was done similarly to $CO_2$ processing, but time lag and spectral corrections were determined from $CO_2$ measurements and applied to COS as recommended by Kohonen et al. (2020).
Gap-filling of the COS flux was done using empirical formulas based on photosynthetically active radiation (PAR) and vapor pressure deficit (VPD), as described by Kohonen et al. (2020). $CO_2$ fluxes were gap-filled and partitioned using a procedure to be explained in more detail in Sect. 2.3.1.

Environmental measurements used in the study include air temperature ($T_a$) at 16.8 m (measured with a Pt100 temperature sensor inside a ventilated custom shield), PAR above the canopy (Li-190SZ quantum sensor, LI-COR, Lincoln, NE, USA),
relative humidity (RH) at 16.8 m height (Rotronic MP102H, Rotronic Instrument Corp., NY, USA), soil temperature ($T_{soil}$) at 2–5 cm depth (KTY81-110 temperature sensor, Philips, The Netherlands) as a mean of five locations and soil water content (SWC) in the humus layer (Delta-T ML2 soil moisture sensor, Delta-T Devices, Cambridge, UK).

## 2.3 GPP calculations

This section describes each of the four methods for estimating GPP. Daily average GPP was only calculated if more than 50%
of the measured 30-min flux data was available for each day, and monthly averages were calculated from the daily means. In Vesala et al. (2022), COS fluxes were found to have 52% data availability on average. While setting a 50% threshold is somewhat subjective, it ensures that the analyzed daily estimates of GPPs reflect measured fluxes rather than the gap-filling procedure. Gap-filled flux data were used in estimating diurnal variation and cumulative GPP. All comparisons between the methods used measured (non gap-filled) data only, when both $CO_2$ and COS flux data were available.

### 2.3.1 GPP from traditional $CO_2$ flux partitioning

NEE was partitioned into respiration (R) and $\text{GPP}_{\text{NLR}}$ as

$$\text{NEE} = \text{R} - \text{GPP}_{\text{NLR}} \tag{1}$$

where $R$ was estimated as in the nighttime method

$$\text{R} = R_C Q_{10}^{T_{sa}/10} \tag{2}$$

where $R_C$ is the respiration at a reference temperature (T=0 °C), $Q_{10}$ is the temperature sensitivity of R and $T_{sa}$ is the arithmetic mean between the air temperature at 16.8 m height and soil temperature at 5 cm depth. Previous studies have shown $T_{sa}$ to be a good choice of respiration driver at Hyytiälä forest (Kolari et al., 2009; Lasslop et al., 2012).

When NEE measurements were not available, the GPP model followed the formula

$$\text{GPP}_{\text{NLR}} = \frac{\alpha PAR + P_{max} - \sqrt{(\alpha PAR + P_{max})^2 - 4\Theta PAR \cdot P_{max}}}{2\Theta} f(T_a) \tag{3}$$

where $\alpha$, $P_{max}$ and $\Theta$ are fitting parameters and $f(T_a)$ is an instantaneous temperature response that brings GPP gradually towards zero at freezing temperatures, given by

$$f(T_a) = \frac{1}{1 + e^{(2(T_0 - T_a))}} \tag{4}$$

where $T_0$=-2°C is the inflection point (Kolari et al., 2014).

Parameters $\alpha$, $P_{max}$ and $R_C$ were estimated for 15-day periods while $Q_{10}$ was estimated from the weighted mean of monthly $Q_{10}$ values from June–August over several years. Weights were the inverse of the confidence interval of each $Q_{10}$ estimate. $\Theta$ was determined as the value that gave the best model fit when the partitioning was run during summer months (June–August) over several years (Kulmala et al., 2019). The parameters of Eq. 3 were estimated from GPP partitioned with the nighttime method in Eq. 1. The modeled NEE from Eqs. (3) and (2) was compared with the measured NEE in Fig. B1a.

### 2.3.2 GPP from artificial neural networks

$\text{GPP}_{\text{ANN}}$ from the data driven model was estimated by applying the $\text{NN}_{\text{C-part}}$ algorithm (Tramontana et al., 2020). $\text{NN}_{\text{C-part}}$ is a customized neural network that emulates the bio-physical processes driving both GPP and R at ecosystem scale and has been applied to several vegetation types distributed globally. The network consists of two subnetworks, which simulate GPP and R, respectively. The two subnetworks are connected in the last node of the overall structure, in which the GPP and R signals are combined to calculate NEE. The GPP subnetwork consists of three layers and estimates the ecosystem-level gross photosynthesis using a light-use efficiency (LUE) approach; in particular, instantaneous LUE is estimated by the first two layers while GPP is calculated as the product between LUE and incoming shortwave radiation in the third layer. $\text{NN}_{\text{C-part}}$ has a hybrid nature and gross photosynthesis is partially constrained by emulating the LUE concept.

Each subnetwork relies on specific predictors. Distinguishing features of this model are a) GPP and R derived by other models are not used, b) functional relationships are derived directly from the data and c) the network's weights are tuned by training the machine learning only on NEE measurements. In this experiment we used the same predictors (VPD, incoming shortwave radiation, potential incoming radiation, $T_a$, $T_{soil}$, SWC, wind speed and wind direction) and network structure as applied by Tramontana et al. (2020). However, to ensure the viability of this method, which is limited by the availability of both predictors and NEE measurements, we set lower requirements for the minimum percentage of measured data for both predictors and half hourly NEE. Moreover, data from all available years were pooled for use in a unique multi-year training process. In particular, we applied the following setting: for each year, less than 55% of predictors were gap-filled and at least 365 half-hourly NEE should be measured for both nighttime and daytime. Despite the high percentage of missing data in observations, gaps had generally short duration with limited effects on the uncertainty of predicted outputs. The final GPP$_{\text{ANN}}$ products were derived by applying trained networks on meteorological inputs, and thus do not include NEE data after network training. The modeled NEE from NN$_{\text{C-part}}$ was compared with the measured NEE in Fig. B1b.

### 2.3.3 GPP from COS flux measurements and an empirical LRU radiation relation

Based on previous soil chamber measurements at Hyytiälä forest it is known that the soil COS flux was -2.7 pmol m$^{-2}$ s$^{-1}$ on average with a variation of only 1 pmol m$^{-2}$ s$^{-1}$ during the growing season and a negligible diurnal variation (Kooijmans et al., 2017; Sun et al., 2018a). The average soil flux was thus first subtracted from the quality filtered and gap-filled COS EC fluxes in order to derive the vegetation contribution to the ecosystem COS exchange. GPP was then calculated from the canopy COS fluxes (FCOS) using the formula (Sandoval-Soto et al., 2005; Blonquist et al., 2011)

$$\text{GPP}_{\text{COS}} = \frac{-FCOS}{\text{LRU}} \frac{[CO_2]_a}{[COS]_a} \tag{5}$$

where $[CO_2]_a$ and $[COS]_a$ denote the atmospheric concentrations of $CO_2$ and COS (in mol m$^{-3}$), respectively, at the EC measurement height, measured by the QCL.

LRU was calculated as a function of PAR (LRU$_{\text{PAR}}$) as described by the empirical equation of Kooijmans et al. (2019):

$$\text{LRU}_{\text{PAR}} = \frac{607.26}{PAR} + 0.57 \tag{6}$$

This LRU equation was derived from field chamber measurements (LRU$_{\text{ch}}$) of pine branch $CO_2$ and COS fluxes with two chambers placed at the top of the canopy in 2017 at the same site and were thus independent from the EC flux measurements (Kooijmans et al., 2019).

### 2.3.4 GPP from COS flux measurements and LRU from stomatal optimization model

Finally, we estimated GPP from Eq. (5) using a new theoretical expression for LRU (LRU$_{\text{CAP}}$) derived from the stomatal optimization model CAP (Dewar et al., 2018). Full details of the derivation are given in Appendix A. The LRU$_{\text{CAP}}$ formulation

was based on the following general expression for LRU given by Eqs. (10-11) of Wohlfahrt et al. (2012):

$$\text{LRU} = \frac{1}{1 - \frac{c_i}{c_a}} \frac{\frac{1}{1.21} + \frac{1}{1.14} \frac{g_s^{COS}}{g_b^{COS}}}{1 + \frac{g_s^{COS}}{g_b^{COS}} + \frac{g_s^{COS}}{g_m^{COS}}} \tag{7}$$

where $g_x^{COS}(x = b, s, m)$ are, respectively, the boundary layer, stomatal and mesophyll conductances for COS, $c_a$ is the atmo-
spheric $CO_2$ molar mixing ratio (mol mol$^{-1}$), $c_i$ is the leaf intercellular $CO_2$ molar mixing ratio (mol mol$^{-1}$), and the numerical
factors 1.21 and 1.14 are the ratios of the conductances of $CO_2$ to COS for stomata and the boundary layer (Wohlfahrt et al.,
2012). If it is assumed that the boundary layer and mesophyll conductances are infinite (as done by Dewar et al. (2018)), Eq.
(7) reduces to

$$\text{LRU} = \frac{1}{1.21} \left( 1 - \frac{c_i}{c_a} \right)^{-1}. \tag{8}$$

An analytical expression for $c_i$ was derived from the stomatal optimization model CAP by Dewar et al. (2018), according to
which stomatal conductance maximise leaf photosynthesis, reflecting a trade-off between stomatal limitations to $CO_2$ diffusion
and non-stomatal limitations (NSLs) to carboxylation capacity. The CAP model predicts the value of $c_i$ as an analytical function
of various environmental and physiological factors. Inserting this function into Eq. (8), $\text{LRU}_{CAP}$ can then be expressed as

$$\text{LRU}_{CAP} = \frac{1}{1.21} \frac{c_a}{c_a - \Gamma^*} \left( 1 + \sqrt{\frac{K_{sl}|\psi_c|}{1.6 g_c VPD}} \sqrt{1 + \frac{2\Gamma^* g_c}{\alpha PAR}} \right), \tag{9}$$

where $\Gamma^*$ is the $CO_2$ photorespiratory compensation point (mol mol$^{-1}$), $K_{sl}$ the soil-to-leaf hydraulic conductance (mol
m$^{-2}$ s$^{-1}$ MPa$^{-1}$), $\psi_c$ is the assumed critical leaf water potential (MPa) at which NSLs reduce photosynthesis to zero, $g_c$
is the carboxylation conductance in the absence of NSLs (mol m$^{-2}$ s$^{-1}$) and $\alpha$ is the photosynthetic quantum yield (mol
mol$^{-1}$) in the absence of NSLs (Duursma et al., 2008; Dewar et al., 2018). While $\Gamma^*$ and $\alpha$ vary seasonally with temperature,
for simplicity we used fixed values representing the growing season averages $50 \times 10^{-6}$ mol mol$^{-1}$ and 0.05 mol mol$^{-1}$,
respectively (Bernacchi et al., 2001; Leverenz and Öquist, 1987; Mäkelä et al., 2008). In addition to PAR (mol m$^{-2}$s$^{-1}$) and
VPD measurements (mol mol$^{-1}$), $\text{LRU}_{CAP}$ requires soil moisture measurements through its dependence on the soil component
of $K_{sl}$. All parameter definitions and values are listed in Table 1.

$\text{LRU}_{CAP}$ is based on a generic physiological model of stomatal function whose predictions have been successfully tested
previously (e.g. Lintunen et al. (2020); Salmon et al. (2020); Dewar et al. (2021); Gimeno et al. (2019)). The model parameters
are all physiologically meaningful, and can be measured independently or obtained from the literature. This formulation there-
fore represents a clear advance on previous COS-based methods based on empirical fitting ($\text{LRU}_{PAR}$), because it provides a
physiological explanation for variations in LRU that may be more robust when extrapolating to other sites.

In addition, $\text{LRU}_{CAP}$ was calculated using a combination of literature values and fitted parameters by fitting the parameter
combinations $X = \frac{|\psi_c|}{1.6 g_c}$ (MPa m$^2$ s mol$^{-1}$) and $Y = \frac{2\Gamma^* g_c}{\alpha}$ (mol m$^{-2}$ s$^{-1}$) to Eq. (9). This analysis was aimed at assessing the
parameter sensitivity of $\text{LRU}_{CAP}$. While the literature-based parameter values gave $X = 2.5$ and $Y = 0.001$, the fitting values

**Table 1.** Explanations, literature values and sources of the parameters used in the $\text{LRU}_{\text{CAP}}$ formulation for Hyytiälä forest. $c_a$ was derived from the measurements in Kooijmans et al. (2019), SWC, PAR and VPD form measurements done in this study. Soil-related values ($K_{soil,sat}$, $r_{cyl}$, $SWC_{sat}$ and $b$) are for soil horizon B (which was considered to be representative of the rooting zone), where the SWC measurements were also made.

| Symbol | Definition | Default value or formula and unit | Source |
|---|---|---|---|
| $c_a$ | Atmospheric $CO_2$ molar mixing ratio | $415\times10^{-6}$ mol mol$^{-1}$ | Measured |
| $\Gamma^*$ | Photorespiratory compensation point of $CO_2$ | $50\times10^{-6}$ mol mol$^{-1}$ | Bernacchi et al. (2001) |
| $g_c$ | Carboxylation conductance in the absence of non-stomatal limitations | 0.5 mol m$^{-2}$ s$^{-1}$ | Dewar et al. (2018) |
| $\psi_c$ | Critical leaf water potential | -2 MPa | Dewar et al. (2018) |
| $\alpha$ | Photosynthetic quantum yield | 0.05 mol mol$^{-1}$ | Leverenz and Öquist (1987) |
| $K_{sl}$ | Leaf-specific soil-to-leaf hydraulic conductance | $\frac{K_{soil}K_x}{K_{soil}+K_x}$ ; mol m$^{-2}$s$^{-1}$MPa$^{-1}$ | |
| $K_x$ | Leaf-specific root-to-leaf xylem hydraulic conductance | $0.78\times10^{-3}$ mol m$^{-2}$s$^{-1}$MPa$^{-1}$ | Duursma et al. (2008) |
| $K_{soil}$ | Leaf-specific soil hydraulic conductance | $\frac{R_1}{\text{LAI}}\frac{2\pi k_{soil}}{\log\left(\frac{r_{cyl}}{r_{root}}\right)}$ ; mol m$^{-2}$s$^{-1}$MPa$^{-1}$ | |
| $k_{soil}$ | Soil hydraulic conductivity | $k_{soil,sat}\left(\frac{\text{SWC}}{SWC_{sat}}\right)^{2b+3}$ ; mol m$^{-1}$s$^{-1}$MPa$^{-1}$ | |
| $k_{soil,sat}$ | Saturated soil hydraulic conductivity | 5.7 mol m$^{-1}$s$^{-1}$MPa$^{-1}$ | Duursma et al. (2008) |
| $R_1$ | Root length index | 5300 m$^{-1}$ | Nikinmaa et al. (2013) |
| LAI | Leaf area index, all-sided | 8 m$^2$m$^{-2}$ | Measured |
| $r_{cyl}$ | Radius of the cylinder of soil accessible to a root | 0.00458 m | Duursma et al. (2008) |
| $r_{root}$ | Fine root radius | $0.3\times10^{-3}$ m | Nikinmaa et al. (2013) |
| $SWC_{sat}$ | Saturation soil water content | 0.52 m$^3$ m$^{-3}$ | Duursma et al. (2008) |
| SWC | Soil water content | m$^3$ m$^{-3}$ | Measured |
| $b$ | Parameter of the soil water retention curve | 4.46 | Duursma et al. (2008) |
| VPD | Vapor pressure deficit | mol mol$^{-1}$ | Measured |
| PAR | Photosynthetically active radiation | mol m$^{-2}$s$^{-1}$ | Measured |

were $X=2.64$ and $Y=0.0033$ and gave a slightly better agreement of $\text{LRU}_{\text{CAP}}$ with $\text{LRU}_{\text{ch}}$ (RMSE=1.89, while without fitting RMSE=2.01, Fig. B2). However, we emphasise that this fitting procedure was conducted purely in order to assess the model performance and is not a requirement for applying $\text{LRU}_{\text{CAP}}$ in practice when literature-based parameter values are available. Moreover, the results presented in this article are not based on fitted parameter values, but on literature values only.

 ## 3 Results and discussion

### 3.1 Environmental conditions

March 2013 was colder than other years (average -7.0 °C), and also had the highest average PAR (207.3 $\mu$mol m$^{-2}$ s$^{-1}$) and lowest soil moisture (0.23 m$^3$ m$^{-3}$) (Fig. 1). A clear increase in VPD and decrease in soil moisture were seen in August 2013, with soil moisture decreasing from 0.24 in July to 0.19 m$^3$ m$^{-3}$ in August and afternoon median VPD increasing to 1.00 kPa.
 July 2014 was warmer (19.0 °C) and dryer (VPD 0.88 kPa) than other years but soil moisture remained high at 0.25 m$^3$ m$^{-3}$. In 2015, VPD increased from 0.44 in July to 0.62 kPa in August and soil moisture decreased from 0.31 in July to 0.24 m$^3$ m$^{-3}$ in August. May 2017 had high amounts of radiation (monthly average PAR of 478.4 $\mu$mol m$^{-2}$ s$^{-1}$) and soil temperature was low (3.4 °C), while soil moisture and VPD were at a normal level at 0.28 m$^3$ m$^{-3}$ and 0.47 kPa, respectively. Soil moisture in September–December in 2017 was 10 % higher than other years, while no significant differences between years were found in
 other environmental variables in late autumn.

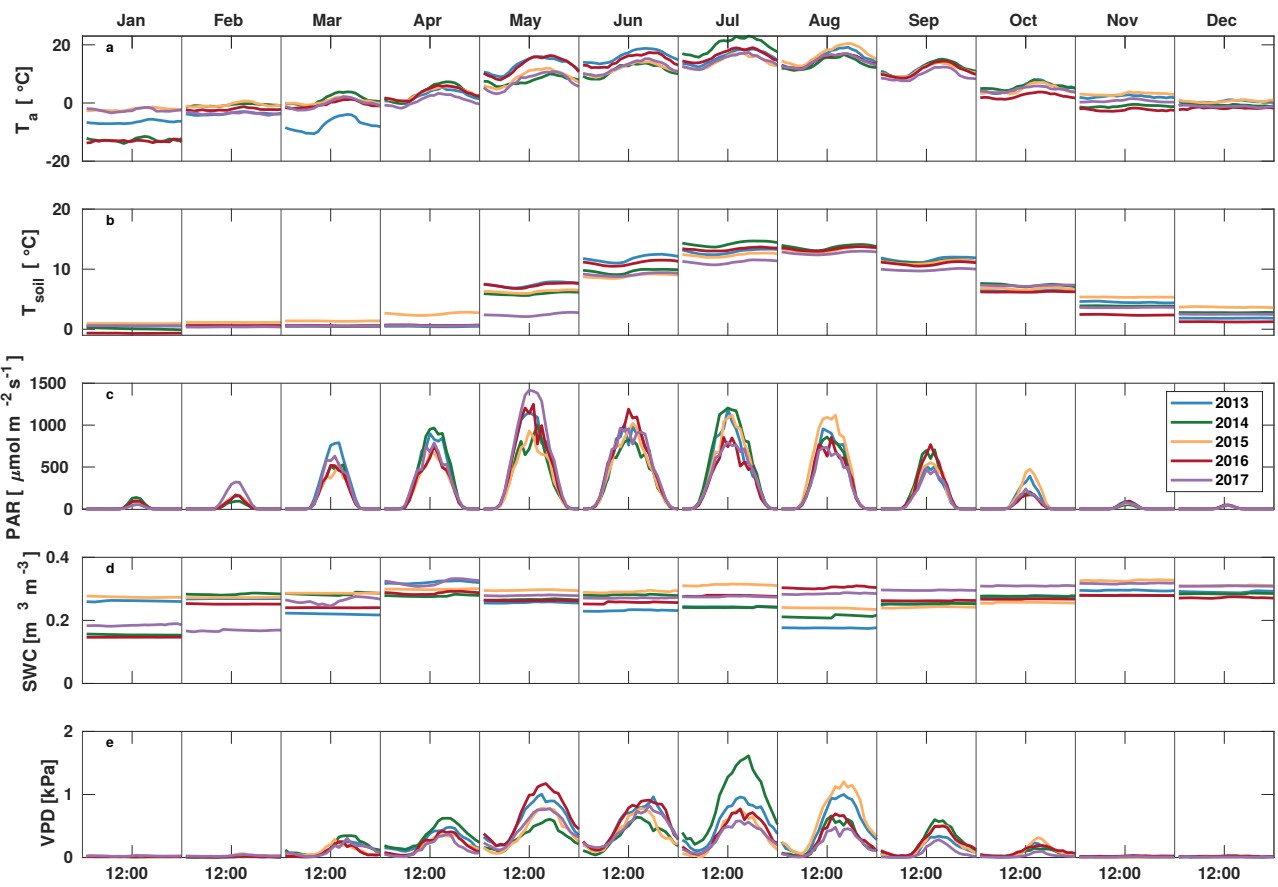

**Figure 1.** Median diurnal variation of $T_a$, $T_{soil}$, PAR, SWC and VPD in different months during the measurement period 2013–2017.

## 3.2 GPP comparison from sub-daily to seasonal scales

Midday $GPP_{ANN}$ was on average 12 % higher than midday $GPP_{NLR}$ during the summer months (May–July) in 2014 and 2017 (Figs. 2,3,4a), opposite to the result found by Tramontana et al. (2020) in a comparison of $GPP_{ANN}$ with standard FLUXNET partitioning during summer months at multiple sites. The difference between $GPP_{NLR}$ and $GPP_{ANN}$ during other months was

220 negligible. We compared the more common use of air temperature as the respiration driver, $GPP_{airT}$, (instead of the average of soil and air temperatures) against $GPP_{NLR}$ and found that the two methods agreed very well with each other at all time scales (Fig. B3). The small differences in the diurnal variations of $GPP_{NLR}$ and $GPP_{ANN}$ are thus not due to the choice of temperature measurement as respiration driver. During the measurement period 2013–2017, 30 min, daily and monthly $GPP_{ANN}$ did not differ statistically from $GPP_{NLR}$ (tested with the ANOVA test; Fig. B4-B6). However, on 30 min time scale the $GPP_{ANN}$ was

225 on average 15 % lower than $GPP_{NLR}$. The lower agreement of 30 min $GPP_{ANN}$ and $GPP_{NLR}$ than on longer time scales may have resulted from the $NN_{C-part}$ method restricting $GPP_{ANN}$ to only positive values while $GPP_{NLR}$ may take on negative values due to random noise in the NEE measurements. The relative and absolute differences of $GPP_{ANN}$ to $GPP_{NLR}$ are, however, very small when averaging over longer time periods (relative difference 2 % on average during summer months, Fig. 5).

$GPP_{COS,PAR}$ was very similar to $GPP_{NLR}$ especially during morning and early evening (Figs. 2 and 3), but showed higher midday values than $GPP_{NLR}$, especially during summer months (May–August) in all years. At the daily scale, $GPP_{COS,PAR}$ was on average 23 % higher than $GPP_{NLR}$ (Figs. 4e and 5) and also differed from $GPP_{NLR}$ and $GPP_{ANN}$ statistically (p<0.01) on 30 min and daily scales (ANOVA test). On monthly scale, there was no statistical difference to any of the other GPP methods.

Based on the CAP stomatal optimisation model, $LRU_{CAP}$ requires PAR, SWC and VPD as well as ecosystem specific literature values for some parameters as input variables. In contrast, $LRU_{PAR}$ by Kooijmans et al. (2019) only uses PAR. $LRU_{CAP}$ therefore takes into account additional effects of drought and air humidity on LRU. In spring, the diurnal variation of $GPP_{COS,CAP}$ closely follows that of $GPP_{NLR}$ and $GPP_{ANN}$ until June (Figs. 2 and 3). Especially in June and July $GPP_{COS,CAP}$ is lower than the other GPP estimates. On 30 min time scale $GPP_{COS,CAP}$ is on average 12 % lower than $GPP_{NLR}$, but there

is large scatter due to noisy FCOS measurements, as for $GPP_{COS,PAR}$. However, there is less scatter in $GPP_{COS,CAP}$ than $GPP_{COS,PAR}$ (Fig. 4d,g), indicating that some of the scatter is due to LRU estimation. On daily scales $GPP_{COS,CAP}$ is 7% higher than $GPP_{NLR}$ and on monthly scales the difference decreases to 3%. However, there is no statistically significant difference between the 30 min and monthly values of $GPP_{NLR}$ and $GPP_{COS,CAP}$ (ANOVA test). The relative and absolute difference between $GPP_{COS,CAP}$ and $GPP_{NLR}$ is also generally smaller than between $GPP_{COS,PAR}$ and $GPP_{NLR}$ throughout

the year (Fig. 5). In addition, $GPP_{COS,CAP}$ reproduces the same two distinctive probability density function peaks as $GPP_{NLR}$ and $GPP_{ANN}$ at 1.7 and 6.6 $\mu$mol m$^{-2}$s$^{-1}$ while $GPP_{COS,PAR}$ finds weaker peaks at 2.4 and 7.4 $\mu$mol m$^{-2}$s$^{-1}$ (Fig. 6). In summary, $GPP_{COS,CAP}$ gives better agreement with traditional $GPP_{NLR}$ partitioning than $GPP_{COS,PAR}$. However, $LRU_{CAP}$ was higher than $LRU_{ch}$ and $LRU_{PAR}$ at high radiation (PAR > 1000 $\mu$mol m$^{-2}$s$^{-1}$, Fig. B7a). This may reflect intrinsic

differences in the dependence of $LRU_{PAR}$ and $LRU_{CAP}$ on environmental drivers (PAR, VPD, SWC), as both estimates of
LRU are based on conditions at the top of the canopy.

$LRU_{CAP}$ was also calculated based on a combination of literature values and the fitted parameters $X$ and $Y$ (Sect. 2.3.4 and A1) in order to assess the sensitivity to parameter values. While literature values gave $X = 2.5$ MPa m$^2$ s mol$^{-1}$ and $Y = 0.001$ mol m$^{-2}$ s$^{-1}$, fitting gave $X = 2.64$ MPa m$^2$ s mol$^{-1}$ and $Y = 0.0033$ mol m$^{-2}$ s$^{-1}$ and a slightly better agreement of $LRU_{CAP}$ with measured LRU (RMSE=1.89, while without fitting RMSE=2.01). Thus, while $X$ was close to its literature
value, $Y$ was estimated three times higher. This mismatch suggests there may be scope for further model improvement, such as the inclusion of dark respiration and/or finite mesophyll and boundary layer conductances in the $LRU_{CAP}$ model. However, as the difference between fitted $LRU_{CAP}$ and literature-based $LRU_{CAP}$ (statistical significance tested with Student's t-test, p<0.01) was not large, with a median difference of 4 %, and the applicability of the model without fitting is better, we decided to use the literature-based parameterisation of $LRU_{CAP}$ in this study, without fitting to $LRU_{ch}$.

$LRU_{CAP}$ was also calculated assuming finite mesophyll conductance as a further comparison (Sect. A2). The agreement of this method was better than assuming infinite mesophyll conductance at high PAR, but worse at low PAR (Fig. B7d), very similar to the results from Maignan et al. (2021), who modelled LRU at Hyytiälä using the ORCHIDEE model. This version of $LRU_{CAP}$ was also fitted to measured LRU in terms of parameters $X$ and $Y$ (Sect. A2) to make the low PAR $LRU_{CAP}$ better, which resulted in $X = 3.45$ and $Y = 0.0057$, both higher than their expected literature values. We thus concluded that
the assumption of infinite $g_m$ gives an estimate that is closest to $LRU_{ch}$, although the assumption in itself is physiologically unrealistic. Kooijmans et al. (2019) found that internal conductance (a combination of mesophyll conductance and biochemical reactions) might limit leaf-scale FCOS during daytime. We find a better agreement of $LRU_{CAP}$ with $LRU_{ch}$ if $g_m$ is assumed infinite, but there is a mismatch at high PAR, supporting the possibility that $g_m$ might indeed be a limiting factor under high radiation. In CAP, infinite or finite $g_m$ represent two contrasting hypotheses, in which NSLs act either entirely on photosynthetic
capacity, or entirely on $g_m$, respectively. In reality, NSLs may act on both photosynthetic capacity and $g_m$, with one or the other effect being dominant depending on environmental conditions. The contrasting abilities of each hypothesis to explain $LRU_{ch}$ at low vs. high light might be explained by a shift in the action of NSLs from the photosynthetic capacity to $g_m$ as light increases. However, verifying this possibility lies beyond the scope of the present study.

We calculated the cumulative GPP estimates over May–July, 13 weeks around the peak growing season for each year.
Cumulative $GPP_{COS,PAR}$ was on average 25 % higher than cumulative $GPP_{NLR}$ in all studied years. This is higher than the 4.3 % difference reported in Spielmann et al. (2019) and 3.5 % agreement reported in Commane et al. (2015). In contrast, cumulative $GPP_{COS,CAP}$ varied from 17 % higher in 2014 to 15 % lower in 2015, and on average was only 3 % higher than cumulative $GPP_{NLR}$. Cumulative $GPP_{ANN}$ varied from 10 % higher in 2014 to 9 % lower in 2016 than $GPP_{NLR}$, and on average was 0.1 % lower than $GPP_{NLR}$. As stated above, overall $GPP_{ANN}$ was closest to $GPP_{NLR}$ out of the three other GPP estimates.
$GPP_{COS,CAP}$ was closer to both of $CO_2$-based GPP estimates than $GPP_{COS,PAR}$. However, at high PAR, $LRU_{COS,CAP}$ was higher than chamber-based measurements, leading to a lower GPP. Nevertheless, no firm conclusions can be drawn here, as the LRU observations only cover measurements at the top of the canopy, and may not reflect LRU over the whole canopy.

**Table 2.** Cumulative GPP (gC m$^{-2}$) over May–July with different GPP estimates. All sums are calculated from same data coverage and the fraction of gap-filled flux data ($CO_2$ flux for GPP$_{NLR}$, COS flux for GPP$_{COS,PAR}$ and GPP$_{COS,CAP}$) is presented in parentheses. GPP$_{ANN}$ does not include gap-filled NEE data, since it is based on meteorological variables. *In 2015, the cumulative sum covers only July.

| Year | GPP$_{NLR}$ | GPP$_{ANN}$ | GPP$_{COS,PAR}$ | GPP$_{COS,CAP}$ |
|------|-------------|-------------|-----------------|-----------------|
| 2013 | 481 (0.16) | 473 | 597 (0.28) | 510 (0.28) |
| 2014 | 294 (0.20) | 324 | 414 (0.24) | 343 (0.24) |
| 2015 | 193 (0.23)* | 188* | 212 (0.31)* | 165 (0.31)* |
| 2016 | 623 (0.40) | 565 | 722 (0.43) | 599 (0.43) |
| 2017 | 387 (0.35) | 399 | 522 (0.34) | 428 (0.34) |

It has been suggested that, due to the Kok effect, leaf respiration is inhibited under radiation (Kok, 1949). This inhibition has been estimated to be approximately 13 % in the evergreen needle-leaf forests during summer (Keenan et al., 2019). Measurements of $CO_2$ isotope fluxes support the conclusion that, due to the Kok effect, GPP from traditional $CO_2$ flux partitioning using the nighttime method is overestimated (Wehr et al., 2016). However, ecosystem respiration at the Hyytiälä forest site is dominated by soil respiration (Ilvesniemi et al., 2009), so that the Kok effect may be of limited importance in this ecosystem (Keenan et al. (2019); Yin et al. (2020)). Reduced leaf respiration under radiation would be visible as a break point around the compensation point with a change of the slope of NEE against radiation. However, such a break point was not detected in our observations, as is demonstrated in Fig. B8. While it is possible that less radiated needles experience less inhibition than well radiated, that cancel out at the ecosystem scale (Wohlfahrt et al., 2005), this test provides some insight to the problem. It is thus not expected that independent GPP estimates in Hyytiälä would necessarily result in lower GPP than the traditional methods. Moreover, Tramontana et al. (2020) showed that uncertainties and biases in NEE (and COS flux) measurements exceed those resulting from the possible Kok effect.

### 3.3 GPP responses to environmental conditions

All four GPP estimates responded similarly to environmental forcing (PAR, $T_a$, VPD) both in spring and summer (Fig. 7). In spring, all GPP estimates increased with increasing radiation levels, while in summer a saturation point was found at PAR>500 $\mu$mol m$^{-2}$s$^{-1}$, that could be linked to VPD limitation on stomatal conductance in the afternoon (Kooijmans et al., 2019). GPP$_{COS,PAR}$ was higher than GPP$_{COS,CAP}$ at PAR>400 $\mu$mol m$^{-2}$s$^{-1}$ while at low PAR values they agreed well with each other both in spring and summer, as well as with GPP$_{NLR}$ and GPP$_{ANN}$. GPP$_{COS,PAR}$ thus has a stronger radiation response than the other GPP estimates, due to lower empirical LRU estimate than LRU$_{CAP}$ at high PAR (Fig. B7). A similar PAR response was found in Spielmann et al. (2019), who studied GPP$_{COS,PAR}$ with a traditional GPP partitioning method in four different sites in Europe. Although GPP$_{COS,CAP}$ agrees well with both GPP$_{NLR}$ and GPP$_{ANN}$ at high PAR, it is likely underestimated due to high LRU$_{CAP}$ at high PAR (Fig. B7).

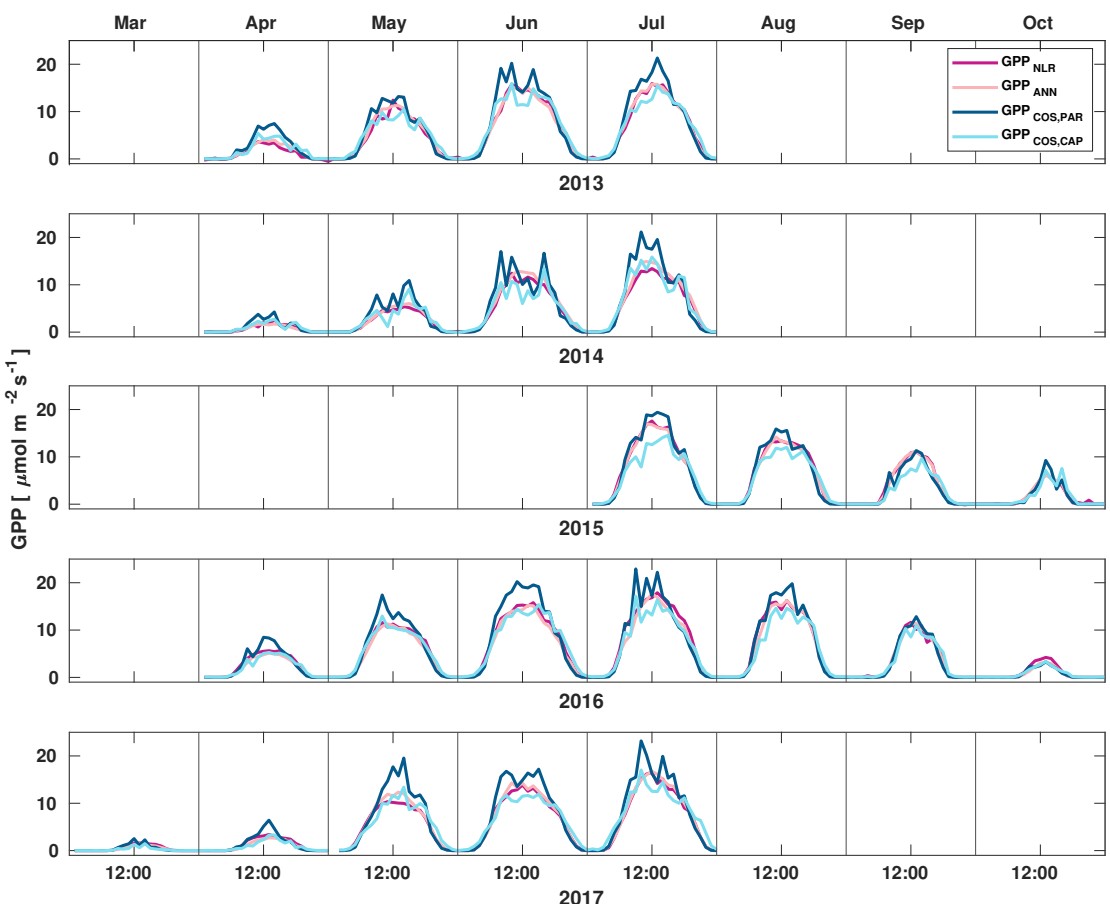

**Figure 2.** Median diurnal variation of GPP partitioned using a combined nighttime-daytime method (GPP$_{NLR}$, purple line), GPP from artificial neural networks (GPP$_{ANN}$, pink line), GPP from COS flux measurements with LRU determined according to Kooijmans et al. (2019) (GPP$_{COS,PAR}$, dark blue line) and GPP from COS flux measurements using a new approach for LRU (Sect. **??**, GPP$_{COS,CAP}$, light blue line) in different months during the measurement period 2013–2017. Averaging was done to the same data points and only months with more than 55 % of data coverage were included.

In spring, increasing air temperature increased all GPP estimates similarly until $T_a$ reached 17°C. However, again GPP$_{COS,PAR}$ was higher than other GPP estimates. In summer, air temperature did not have a notable effect on any GPP estimate. Responses to VPD were similar for each GPP estimates both in spring and summer. In spring, decreasing air humidity (increasing VPD) was associated with increased GPP until VPD>0.7 kPa after which VPD had little or no effect. The apparent increase in GPP with VPD in spring may be caused by the correlation of $T_a$ with VPD, coinciding with the start of the growing season, as the trees are not water-limited after snow melt. In summer, dryness started to limit GPP at VPD>1 kPa. We found that similar to

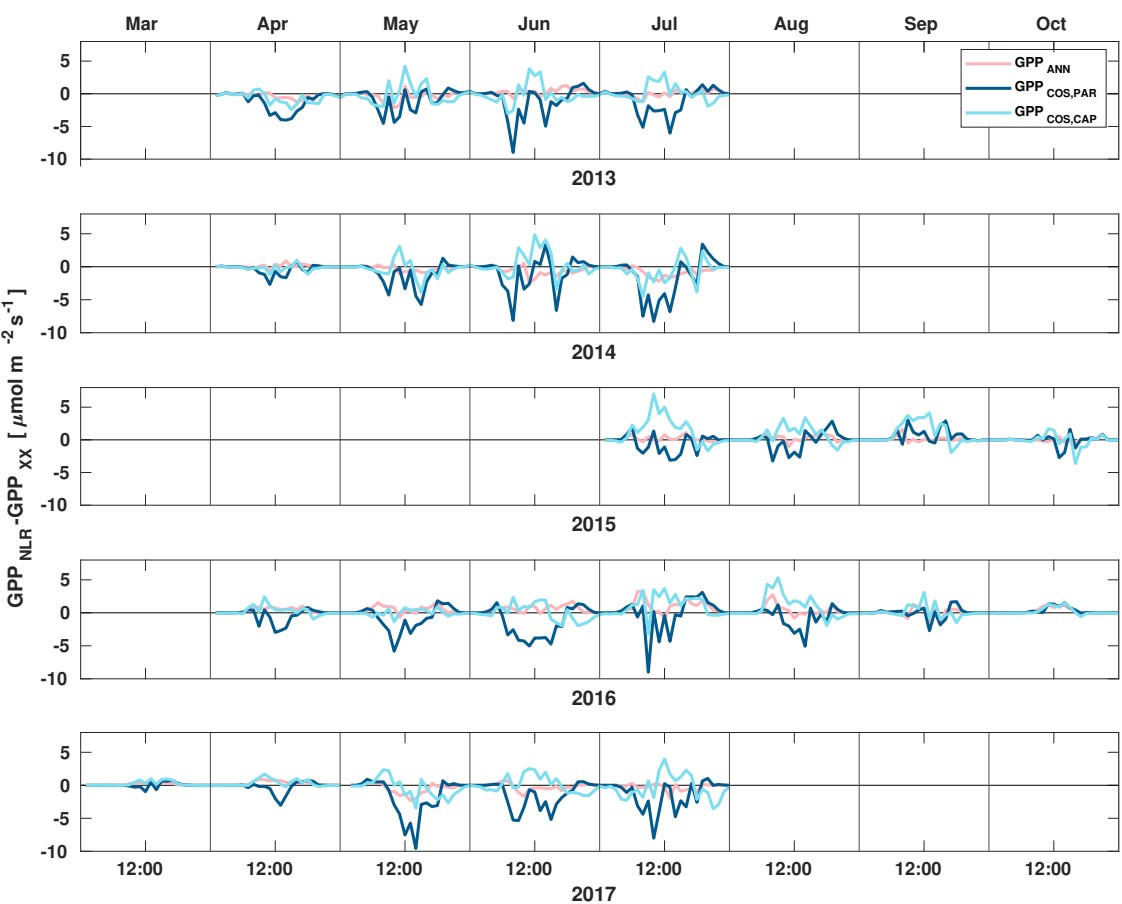

**Figure 3.** Diurnal variation of the difference of GPP$_{\text{ANN}}$ (pink), GPP$_{\text{COS,PAR}}$ (dark blue) and GPP$_{\text{COS,CAP}}$ (light blue) to the reference GPP$_{\text{NLR}}$ in different months during the measurement period 2013–2017. Averaging was done to the same data points and only months with more than 55 % of data coverage were included.

PAR and $T_a$ responses, GPP$_{\text{COS,PAR}}$ was higher than other GPP estimates at low VPD values, but decreased to similar levels at high VPD (1 kPa) both in spring and summer. GPP$_{\text{COS,PAR}}$ gives higher GPP at low VPD than the $CO_2$-based methods, as does GPP$_{\text{COS,CAP}}$ in spring (Fig. 7). This may indicate that some factor is limiting the photosynthesis reaction (e.g. biochemical limitations in $CO_2$ assimilation) even though the diffusion into the leaf is not limited.

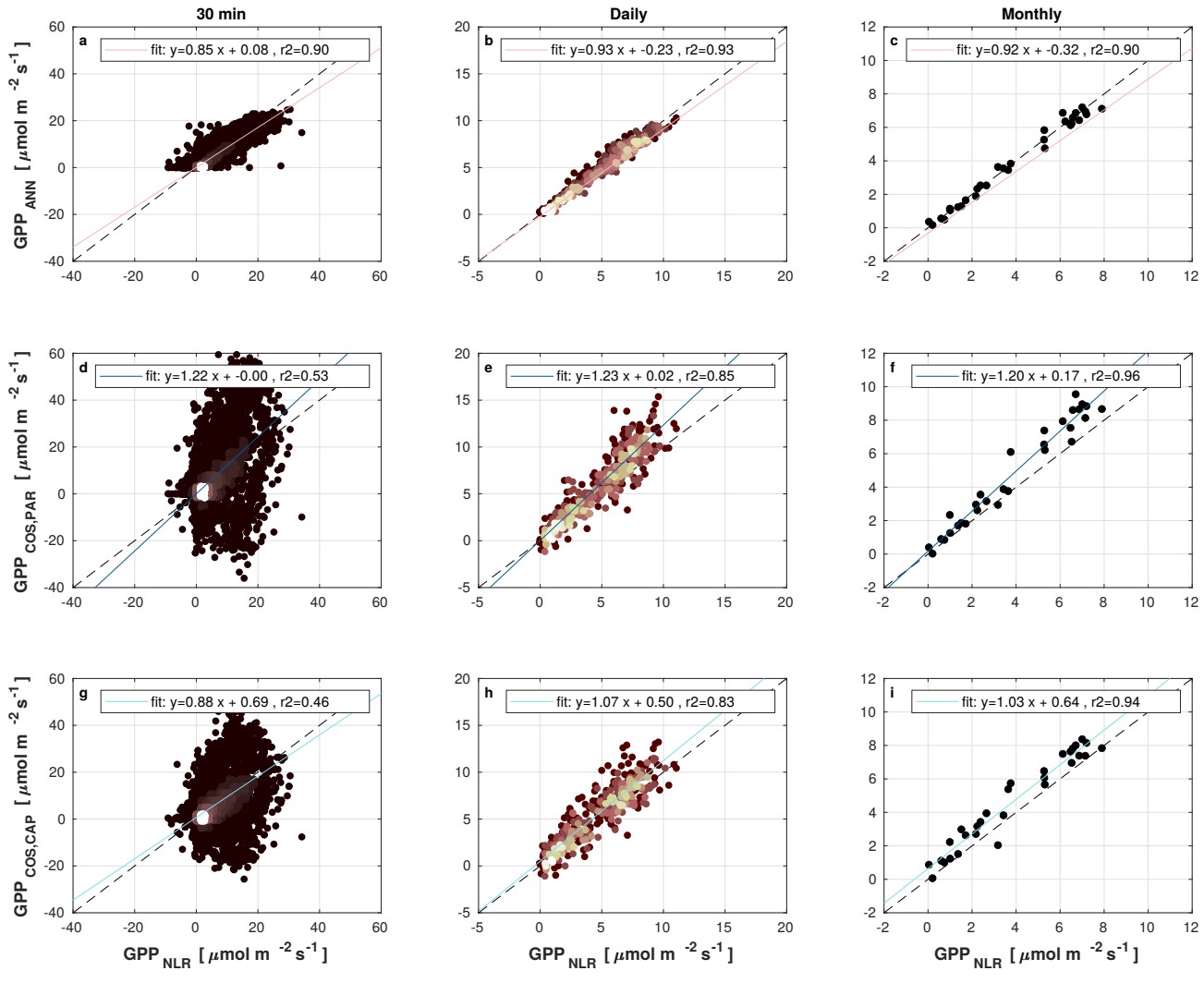

**Figure 4.** Scatter plots of GPP$_{ANN}$, GPP$_{COS,PAR}$ and GPP$_{COS,CAP}$ against GPP$_{NLR}$ in 30 min, daily and monthly time scales. The color of data points in 30 min and daily scatter plots indicate the data density, lighter colors indicating higher point density than dark.

## 3.4 Uncertainties and limitations of the GPP methods

Because ANN fitting is purely based on the provided examples, GPP$_{ANN}$ could be more sensitive to the uncertainty of (training) data with respect to the parametric partitioning methods. Moreover, it is sensitive to missing data especially in the case of long data gaps (Tramontana et al., 2020). The method also requires large data sets for training NN$_{C-part}$, which may not be available

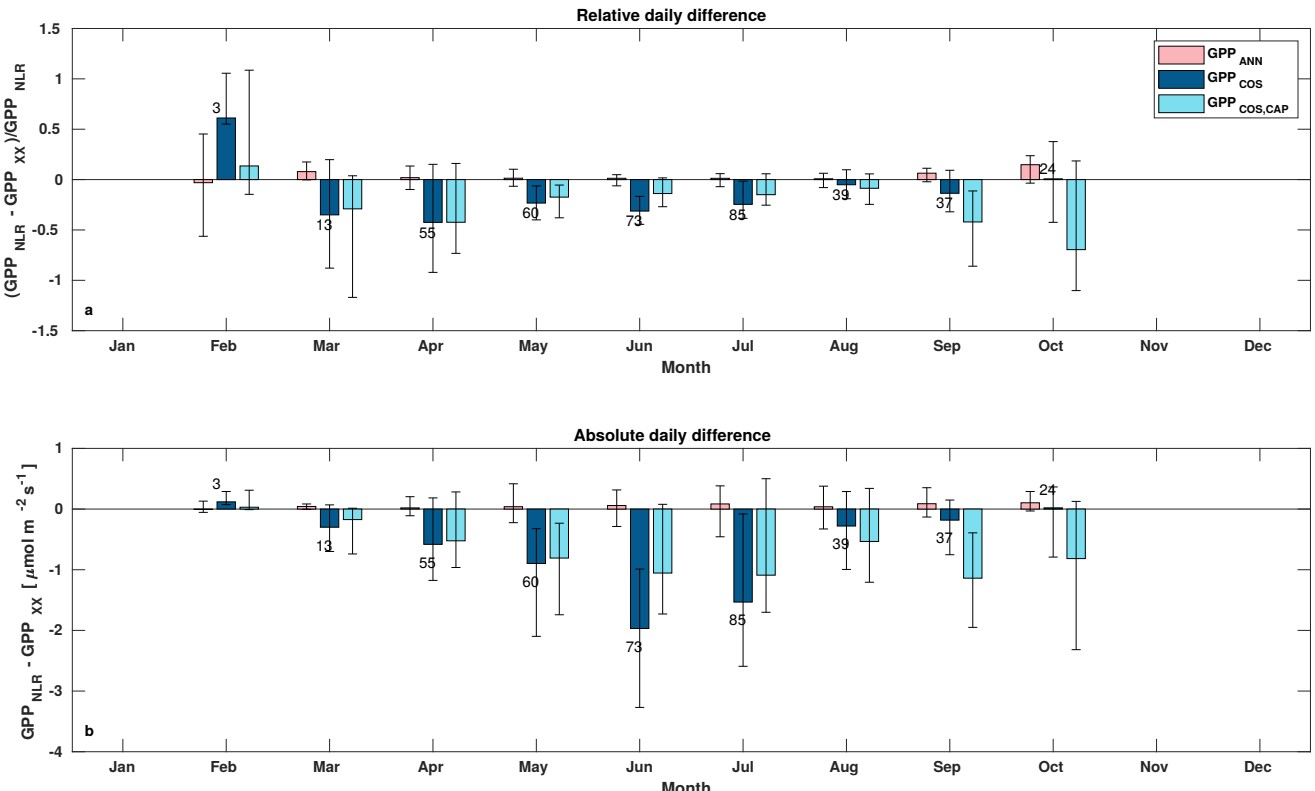

**Figure 5.** Relative (a) and absolute (b) difference of daily GPP$_{\text{ANN}}$ (pink), GPP$_{\text{COS,PAR}}$ (dark blue) and GPP$_{\text{COS,CAP}}$ (light blue) to GPP$_{\text{NLR}}$ in different months, averaged over the whole measurement period 2013–2017. Bars represent the median difference, and whiskers show the 25th and 75th percentiles. Numbers on top of the bars indicate how many daily flux data points have been used for calculating the medians. Median differences have been calculated using the same number of data points for each method in each month.

at all measurement sites. However, GPP$_{\text{ANN}}$ does not require prescribed relationships of GPP to environmental data making it an attractive method for sites with good data availability.

     GPP$_{\text{COS,PAR}}$ uses an empirical PAR relation that is based on measurements at Hyytiälä forest. This PAR relation is site specific and different compared to the one found by Yang et al. (2018). For this reason it is not known if and how it can be used in other sites, where it is suggested to retrieve it directly from observations. The choice of empirical LRU-PAR relation at any

given site is to some extent arbitrary. While the LRU$_{\text{PAR}}$ function is simple, and thereby attractive, it does not take into account the different light conditions inside the canopy, stomatal regulation during drought, or the effects of non-stomatal limitations on photosynthesis. Moreover, being an empirical model, it does not provide a process-based understanding for LRU. While the results of GPP$_{\text{COS,PAR}}$ are promising, we found a 25% difference in midday GPP during summer, similar to that found by Kooijmans et al. (2019). We did not find as good agreement with $CO_2$-based GPP estimates as Asaf et al. (2013), who

found an agreement within 15% using a constant LRU of 1.6 in Mediterranean pine forests and crop fields. However, they also

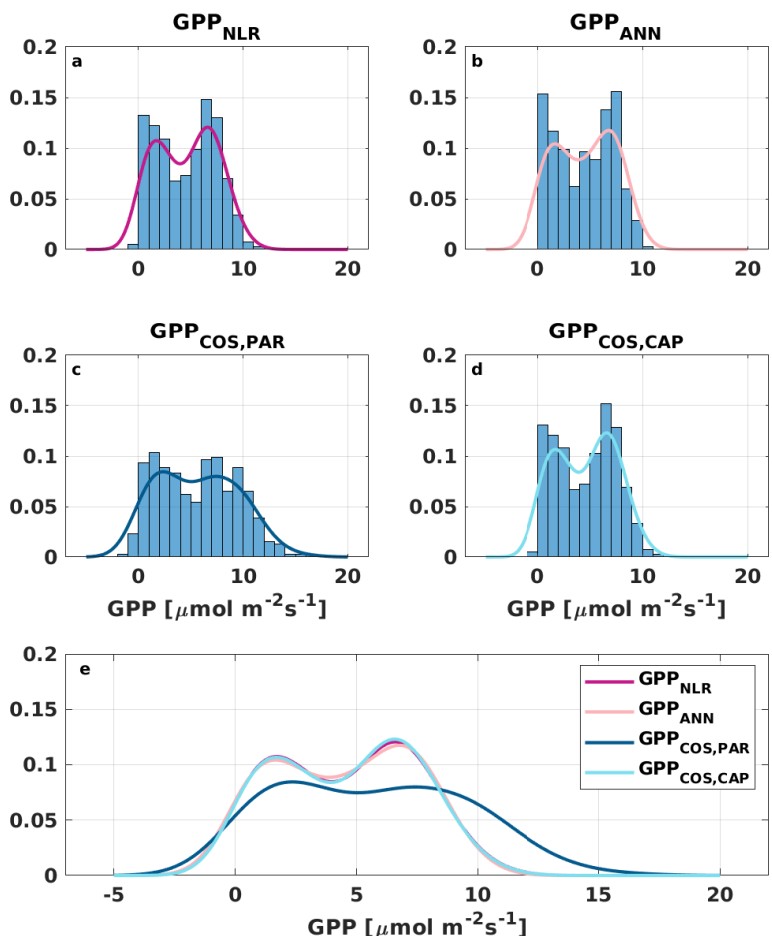

**Figure 6.** Distribution (bars) and probability density functions (lines) of daily average (a) $GPP_{NLR}$, (b) $GPP_{ANN}$, (c) $GPP_{COS,PAR}$ and (d) $GPP_{COS,CAP}$. All probability density functions are combined in (e) for better comparison.

reported higher $GPP_{COS}$ assumed to be related to soil COS uptake, which was not measured or taken into account in their GPP calculations. In our study, we subtracted an average measured soil flux (Sun et al., 2018a) from the ecosystem COS uptake. As the diurnal variation in soil COS exchange was small (less than 1 pmol m$^{-2}$s$^{-1}$) throughout the season, averaging did not make a large difference, and thus soil does not explain the differences found here. However, as soil COS flux measurements are not necessarily available at all sites, this may be one source of uncertainty in wider application. Yang et al. (2018) studied COS flux components and $GPP_{COS}$ in a Mediterranean citrus orchard and found $GPP_{COS}$ to be on average 7 % lower than traditionally partitioned GPP. They also presented a light-dependent and seasonally varying LRU which, however, could not be

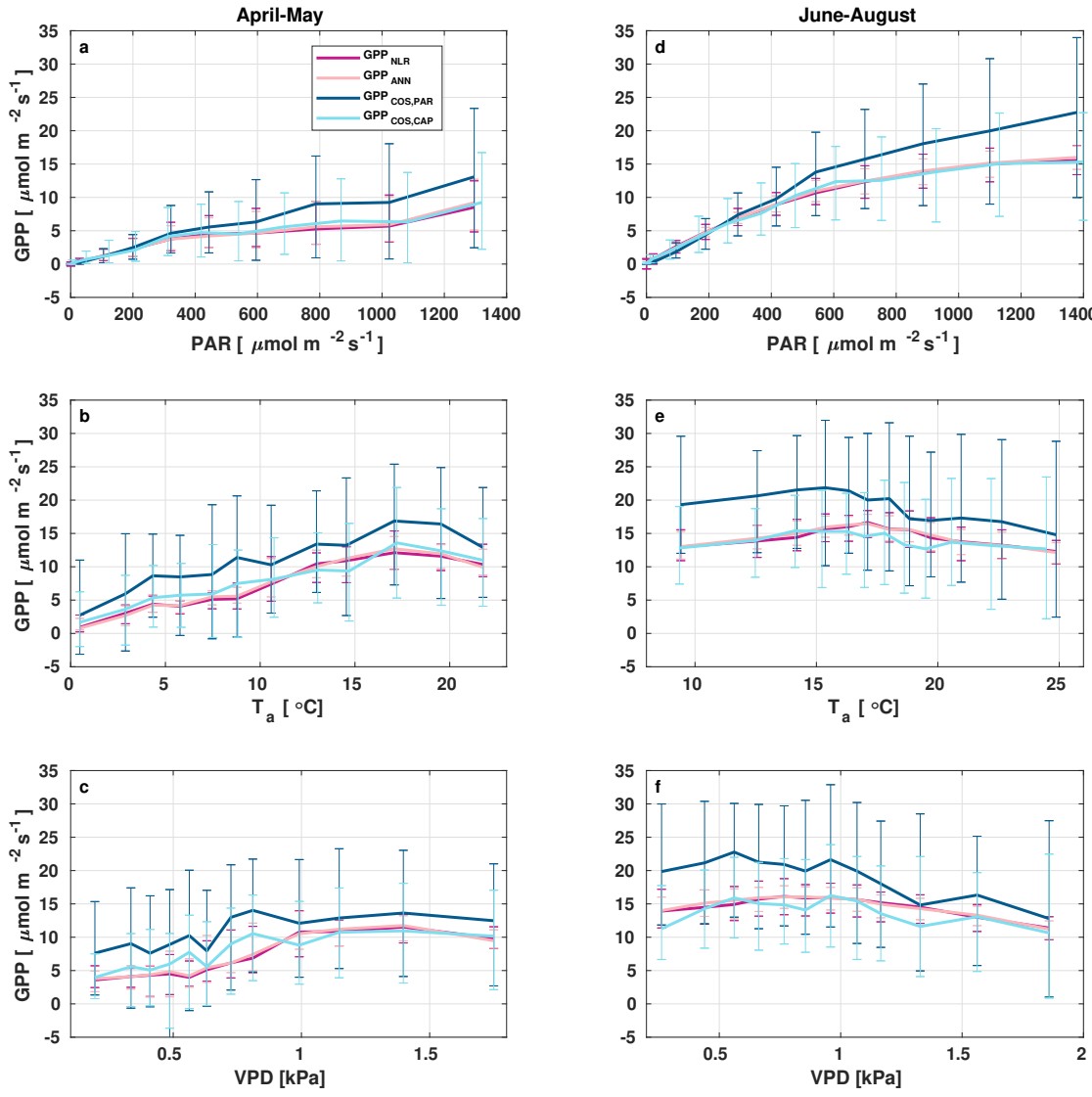

**Figure 7.** Responses of the different GPP estimates (GPP$_{NLR}$ (purple), GPP$_{ANN}$ (pink), GPP$_{COS,PAR}$ (dark blue) and GPP$_{COS,CAP}$, light blue) to environmental parameters – photosynthetically active radiation (a,d), air temperature (b,e) and vapor pressure deficit (c,f) – in spring (a–c) and summer (d–f). Data are binned to 12 equal sized bins (same number of data points in each bin) and all GPPs have the same data coverage. Only measured (non-gap-filled) 30 min flux data was used and GPP was filtered to include only PAR>700 $\mu$mol m$^2$s$^{-1}$ in responses to $T_a$ and VPD to avoid simultaneous correlation with PAR.

applied to Hyytiälä COS fluxes due to the very different ecosystem types studied, indicating that the PAR responses may differ between ecosystems.

$GPP_{COS,CAP}$ may be more applicable at other sites than $GPP_{COS,PAR}$ because it is based on a generic physiological model of stomatal behaviour, which requires only literature-based parameter values and simple meteorological variables as inputs. However, as for $LRU_{PAR}$, $LRU_{CAP}$ should also be tested at other sites against measured LRU, to verify its applicability at other ecosystems. The version of $LRU_{CAP}$ assuming infinite mesophyll conductance, while giving reasonable results in comparison with $LRU_{ch}$, is clearly physiologically unrealistic. The formulation of $LRU_{CAP}$ with finite $g_m$ did not compare as well with

$LRU_{ch}$ at Hyytiälä forest (RMSE=2.58, median difference to $LRU_{ch}$ 22%), especially during low light conditions, but may compare better at other measurement sites.

One source of uncertainty in GPP estimates based on $LRU_{CAP}$ and $LRU_{PAR}$ is that both LRU predictions are calculated from radiation measurements at the top of the canopy, where there is no shading by foliage, although the theoretical dependence of $LRU_{CAP}$ on radiation is more generally applicable throughout the canopy. The branch chamber measurements (on which

the empirical $LRU_{PAR}$ function is based) were also made at the top of the canopy. The measured needles were thus well-adjusted to high radiation conditions. Therefore, we did not take into account light penetration and scattering through canopy. However, the needles and leaves within the canopy are also well adjusted to low light conditions and may be more efficient with their stomatal control in varying light conditions than needles on top of the canopy. Thus, this may not be a large source of uncertainty. However, it is also possible that LRU varies throughout the canopy due to different light conditions.

**4    Conclusions**

Daily $GPP_{ANN}$ and $GPP_{NLR}$ did not differ significantly, and differences were also small on sub-daily and seasonal time scales. $GPP_{COS,PAR}$ was higher than $GPP_{NLR}$ on all time scales studied, including the estimate of three-month cumulative GPP during the peak growing season. In contrast, $GPP_{COS,CAP}$, a new method based on stomatal optimization theory, gave better agreement with $GPP_{NLR}$ on all time scales, and was also less scattered than $GPP_{COS,PAR}$ on a 30-min time scale.

The $LRU_{CAP}$ function provides a new theoretical underpinning for COS-based GPP estimates that can be used at other measurement sites, potentially without requiring additional branch chamber measurements. $LRU_{CAP}$ represents a significant improvement on previous LRU functions based on site-specific empirical regressions. However, $LRU_{CAP}$ overestimated LRU at high radiation, when compared to LRU observations at the top of the canopy, leading to a lower midday $GPP_{COS,CAP}$, especially in summer. This discrepancy may result from the assumption of infinite mesophyll conductance, or the absence of dark

respiration, in the underlying stomatal optimization model. $LRU_{CAP}$ would benefit from further testing at other measurement sites with COS and $CO_2$ branch flux measurements, including measurements inside the canopy for better canopy-integrated LRU estimates.

Although COS flux measurements are noisier, more expensive and more difficult than those of $CO_2$, they provide an opportunity for better process-based understanding of photosynthesis, in comparison with more traditional $CO_2$-based estimates of

GPP. In addition to COS, other proxies such as solar induced fluorescence and isotopic flux measurements should be tested

simultaneously to properly investigate their deficiencies and advantages in estimating GPP and processes underlying photosynthesis.

The establishment of large long-term ecosystem Research Infrastructures (e.g. ICOS, NEON, TERN, see Papale (2020)) – involving sites equipped with eddy covariance systems that could potentially also host COS, SIF and isotope sensors – together with the planned launch of the FLEX satellite in 2025 (https://earth.esa.int/eogateway/missions/flex) that will provide global vegetation fluorescence measurements, open up a new phase in monitoring and understanding plant photosynthesis. Our results also underline the important role of small-scale ecophysiological measurements and models in underpinning these larger-scale initiatives.

*Data availability.* The flux data and all GPP estimates used in this study are available from https://doi.org/10.5281/zenodo.6940750 (Kohonen et al., 2022). Environmental data used in the study are available from http://urn.fi/urn:nbn:fi:att:a8e81c0e-2838-4df4-9589-74a4240138f8 (Aalto et al., 2019). The most recent version of the data is available from https://smear.avaa.csc.fi (last access: 9 June 2020).

## Appendix A: LRU predicted by the CAP stomatal optimization model

### A1 $LRU_{CAP}$ assuming infinite mesophyll conductance

The general expression for the leaf relative uptake ratio (LRU) derived from the diffusion laws for COS and $CO_2$ (Wohlfahrt et al., 2012) is

$$\text{LRU} = \frac{1}{1 - \frac{c_i}{c_a}} \frac{\frac{1}{1.21} + \frac{1}{1.14} \frac{g_s^{COS}}{g_b^{COS}}}{1 + \frac{g_s^{COS}}{g_b^{COS}} + \frac{g_s^{COS}}{g_m^{COS}}}. \tag{A1}$$

where $g_x^{COS}(x = b, s, m)$ are the boundary layer, stomatal and mesophyll conductances for COS, respectively, $c_a$ and $c_i$ are the atmospheric and leaf intercellular $CO_2$ molar mixing ratios (mol mol$^{-1}$), respectively, and the numerical factors 1.21 and 1.14 are the ratios of the conductances of $CO_2$ to COS for stomata and the boundary layer, respectively.

If it is assumed that boundary layer and mesophyll conductances are infinite, Eq. (A1) reduces to

$$\text{LRU} = \frac{1}{1.21} \left( 1 - \frac{c_i}{c_a} \right)^{-1} \tag{A2}$$

We derived $c_i$ from the CAP stomatal optimization model (Dewar et al., 2018), according to which stomatal conductance adjusts to maximise the rate of leaf photosynthesis ($A$) through a trade-off between stomatal and non-stomatal limitations. Our photosynthesis model is based on that of Thornley and Johnson (1990) (their Eq. 9.12i), modified to include non-stomatal limitations (NSLs):

$$A = \left( 1 - \frac{\psi_{leaf}}{\psi_c} \right) \frac{\alpha Q g_c (c_i - \Gamma^*)}{\alpha Q + g_c (c_i + \Gamma^*)}, \tag{A3}$$

where $\alpha$ is the photosynthetic quantum yield (mol mol$^{-1}$) in the absence of NSLs, $Q$ (mol m$^{-2}$s$^{-1}$) is photosynthetically active radiation (PAR), $g_c$ (mol m$^{-2}$ s$^{-1}$) is the initial slope of the $A - c_i$ reponse curve in the absence of NSLs, $\Gamma^*$ (mol mol$^{-1}$) is the photorespiratory $CO_2$ compensation point, $\psi_{leaf}$ (MPa) is the leaf water potential and $\psi_c$ (MPa) is the critical leaf water potential at which NSLs reduce photosynthesis to zero. In Eq. (A3), NSLs are represented as an apparent downregulation of the $A - c_i$ response curve by a factor that decreases with decreasing leaf water potential, as has been observed in numerous experiments (e.g. Lintunen et al. (2020); Salmon et al. (2020)). Consequently, as stomatal conductance increases there is a trade-off between increased $CO_2$ supply and increased NSLs, such that $A$ has a maximum at some optimal value of stomatal conductance.

We used Eq. (A3) rather than the Farquhar photosynthesis model (Farquhar et al., 1980) because, in the latter, the abrupt switch from Rubisco- to electron transport limitation introduces artificial discontinuities in the CAP solution for optimal stomatal conductance (Dewar et al., 2018), whereas in Eq. (A3) there is a smooth transition from $CO_2$- to light limitation and no such discontinuities occur. The parameter $g_c$ is equivalent to $V_{cmax}/(k_m + \Gamma^*)$ in the Farquhar model.

The CAP solution for optimal stomatal conductance (Dewar et al., 2018) predicts that

$$\frac{c_i - \Gamma^*}{c_a - \Gamma^*} = \frac{1}{1 + \beta}, \tag{A4}$$

where

$$\beta = \sqrt{\frac{1.6D}{K_{sl}|\psi_c|}\left(\frac{1}{g_c}+\frac{2\Gamma^*}{\alpha Q}\right)^{-1}}, \tag{A5}$$

in which $D$ is vapor pressure deficit (VPD; mol mol$^{-1}$) and $K_{sl}$ is the leaf-specific soil-to-leaf hydraulic conductance (mol m$^{-2}$s$^{-1}$MPa$^{-1}$). Writing Eq. (A2) in the equivalent form

$$\text{LRU} = \frac{1}{1.21}\frac{c_a}{c_a-\Gamma^*}\left(1-\frac{c_i-\Gamma^*}{c_a-\Gamma^*}\right)^{-1} \tag{A6}$$

and substituting the CAP prediction from Eqs. (A4) and (A5) then gives

$$\text{LRU}_{\text{CAP}} = \frac{1}{1.21}\frac{c_a}{c_a-\Gamma^*}\left(1+\sqrt{\frac{K_{sl}|\psi_c|}{1.6Dg_c}}\sqrt{1+\frac{2\Gamma^*g_c}{\alpha Q}}\right) \tag{A7}$$

In Eq. (A7) all the parameters are physiologically meaningful and can be measured independently or obtained from the literature, because the underlying CAP model is based entirely on such parameters. This contrasts with use of the stomatal optimization model of Medlyn et al. (2011), for example, which contains an undetermined parameter ($\lambda$, interpreted as the marginal water cost of carbon gain) that must be empirically fitted.

Nevertheless, to assess the performance of LRU$_{\text{CAP}}$ obtained from literature-based parameter values, we compared it with LRU$_{\text{CAP}}$ obtained by fitting the two key parameter combinations $X = \frac{|\psi_c|}{1.6g_c}$ and $Y = \frac{2\Gamma^*g_c}{\alpha}$ in terms of which Eq. (A7) may be written as

$$\text{LRU}_{\text{CAP}} = \frac{1}{1.21}\frac{c_a}{c_a-\Gamma^*}\left(1+\sqrt{\frac{K_{sl}X}{D}}\sqrt{1+\frac{Y}{Q}}\right). \tag{A8}$$

Parameters $X$ and $Y$ were optimized to minimize the RMSE of log(LRU$_{\text{CAP}}$) to measured log(LRU), due to the logarithmic nature of LRU, with Matlab's $fminsearch$ function. However, we emphasise that this fitting procedure was conducted purely in order to assess the model performance and is not a requirement for applying LRU$_{\text{CAP}}$ in practice when literature-based parameter values are available. Moreover, the results presented in this study are not based on the optimized values, but on literature values only.

## A2   LRU$_{\text{CAP}}$ assuming finite mesophyll conductance

In the case that mesophyll conductance is not assumed infinite (but boundary layer conductance is infinite), Eq. (A1) becomes

$$\text{LRU} = \frac{1}{1.21}\frac{1}{1+\frac{g_s^{COS}}{g_m^{COS}}}\left(1-\frac{c_i}{c_a}\right)^{-1}. \tag{A9}$$

If we further assume that the ratios of stomatal to mesophyll conductances are the same for $CO_2$ and COS, then from $g_s^{CO2}(c_a-c_i)=g_m^{CO2}(c_i-c_c)$, where $c_c$ is the chloroplast $CO_2$ molar mixing ratio (mol mol$^{-1}$), we can make the substitution

$$\frac{g_s^{COS}}{g_m^{COS}}=\frac{g_s^{CO2}}{g_m^{CO2}}=\frac{c_i-c_c}{c_a-c_i}, \tag{A10}$$

in Eq. (A9) to obtain

$$\text{LRU} = \frac{1}{1.21}\left(1 - \frac{c_c}{c_a}\right)^{-1}, \tag{A11}$$

which reduces to Eq. (A2) when mesophyll conductance is infinite (since then $c_c = c_i$). As noted above, CAP represents NSLs in terms of an apparent downregulation of the $A - c_i$ response curve (Eq. A3). This empirical observation may be interpreted in various ways: as a downregulation of photosynthetic efficiencies ($\alpha$ and $g_c$) in the chloroplast, or a downregulation of mesophyll conductance ($g_m^{CO2}$), or some combination of the two. In the case where NSLs act entirely on $g_m^{COS}$ with no effect on the biochemical efficiencies, $A$ is given by as a function of the chloroplast $CO_2$ concentration by (cf. Eq. A3)

$$A = \frac{\alpha Q g_c (c_c - \Gamma^*)}{\alpha Q + g_c (c_c + \Gamma^*)}. \tag{A12}$$

In this case, since Eq. (A3) still holds, we obtain the same optimal CAP solution for stomatal conductance and $c_i$ (Eq. A4) as before, but now with an additional prediction for the finite (but variable) mesophyll conductance as implied by Eq. (A12), which links the chloroplast $CO_2$ concentration ($c_c$) to the CAP solution of $A$. From Eq. (A12),

$$c_c - \Gamma^* = \frac{\left(\frac{\alpha Q}{g_c} + 2\Gamma^*\right) A}{\alpha Q - A}. \tag{A13}$$

The CAP solution for stomatal conductance is given by (Dewar et al., 2018)

$$g_s = \frac{\alpha Q}{\frac{\alpha Q}{g_c} + 2\Gamma^*} \frac{x\theta}{x\beta^2 + (1-x)(xw+1)} \tag{A14}$$

where

$$\theta = 1 - \frac{\psi_{soil}}{\psi_c} \tag{A15}$$

$$w = \frac{c_a - \Gamma^*}{\frac{\alpha Q}{g_c} + 2\Gamma^*} \tag{A16}$$

$$x = \frac{c_i - \Gamma^*}{c_a - \Gamma^*} = \frac{1}{1 + \beta} \tag{A17}$$

in which $\psi_{soil}$ (MPa) is the soil water potential. Substituting $x$ as a function of $\beta$ into Eq. (A14) and simplifying gives

$$g_s = \frac{\alpha Q}{\frac{\alpha Q}{g_c} + 2\Gamma^*} \frac{\theta}{\beta(1 + \beta + \frac{w}{1+\beta})}. \tag{A18}$$

We then find the CAP solution for $A$ as follows:

$$A = g_s(c_a - c_i)$$

$$= g_s(c_a - \Gamma^*)(1 - x)$$

$$= g_s(c_a - \Gamma^*)\frac{\beta}{1 + \beta}$$

$$= \frac{\alpha Q(c_a - \Gamma^*)}{\frac{\alpha Q}{g_c} + 2\Gamma^*} \frac{\theta}{(1 + \beta)^2 + w}. \tag{A19}$$

Substituting this into Eq. (A13) and simplifying then gives

$$
\begin{aligned}
\frac{c_c - \Gamma^*}{c_a - \Gamma^*} &= \frac{\theta}{(1+\beta)^2 + (1-\theta)w} \\
&= \frac{\theta}{(1+\beta)^2 + (1-\theta)\frac{g_c(c_a - \Gamma^*)}{\alpha Q + 2\Gamma^* g_c}}
\end{aligned}
\tag{A20}
$$

which can be combined with Eq. (A11) to give the solution of $\mathrm{LRU}_{\mathrm{CAP}}$ with finite mesophyll conductance.

As for $\mathrm{LRU}_{\mathrm{CAP}}$ with infinite mesophyll conductance, we also fitted this version with respect to parameters $X$ and $Y$ in order to compare with the performance of the model using literature-based values. For this procedure, $\beta$ and $w$ were expressed in terms of $X$ and $Y$,

$$
\beta = \frac{1}{\sqrt{\frac{K_{sl}X}{D}\left(1 + \frac{Y}{Q}\right)}}
\tag{A21}
$$

and

$$
w = \frac{c_a - \Gamma^*}{2\Gamma^*} \frac{1}{\frac{Q}{Y} + 1}
\tag{A22}
$$

and then substituted into Eq. (A20). However, as for the infinite $g_m$ solution, this fitting procedure was conducted purely in order to assess the model performance and is not a requirement for applying $\mathrm{LRU}_{\mathrm{CAP}}$ in practice when literature-based parameter values are available.

## Appendix B: Additional figures

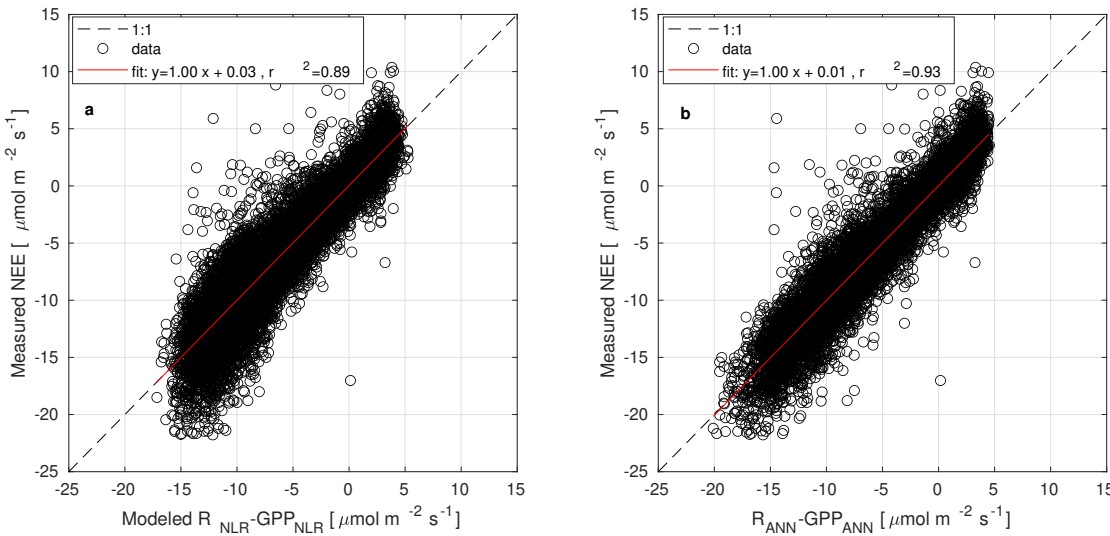

**Figure B1.** Modeled against measured NEE using (a) NLR and (b) ANN models for modeling NEE.

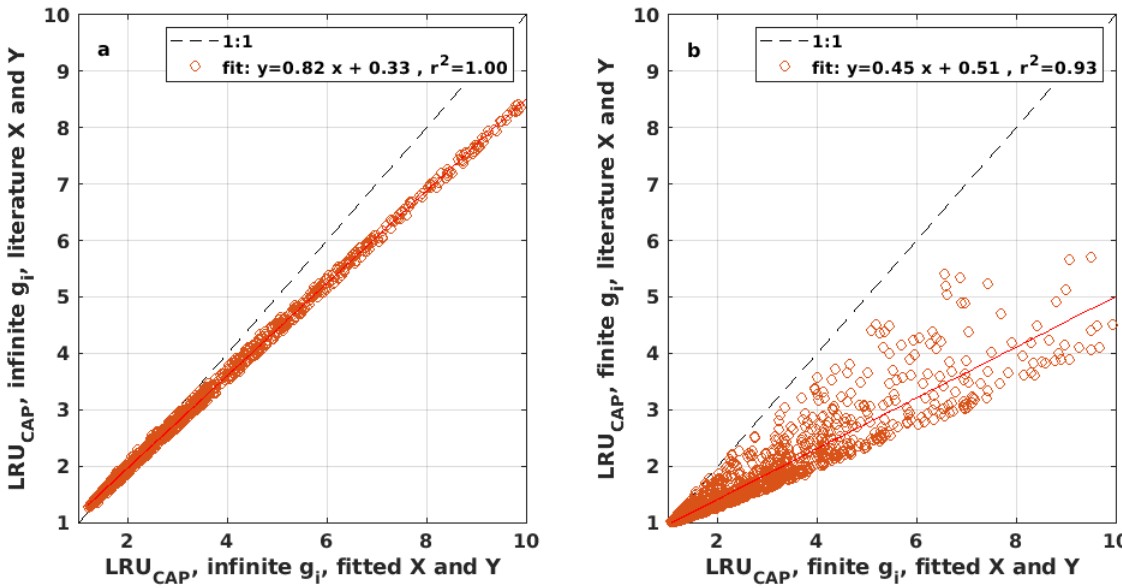

**Figure B2.** Scatter plots of LRU$_{\text{CAP}}$ using the literature values against LRU$_{\text{CAP}}$ using the optimized parameter values when assuming (a) infinite or (b) finite mesophyll conductance.

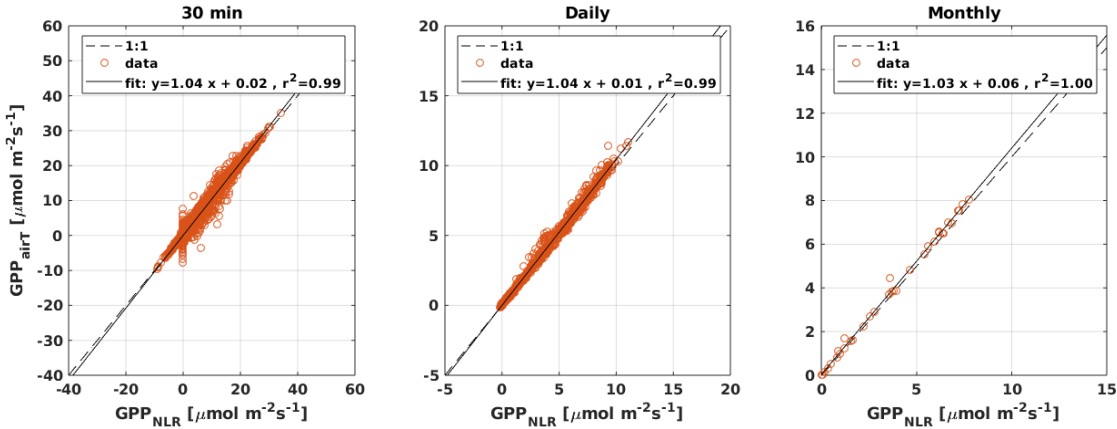

**Figure B3.** Scatter plots of GPP$_{airT}$ that uses only air temperature as the driver for respiration against GPP$_{NLR}$ that uses an average of air and soil temperatures as the respiration driver on 30 min, daily and monthly time scales. Black solid line is the least-squares linear fit to the data.

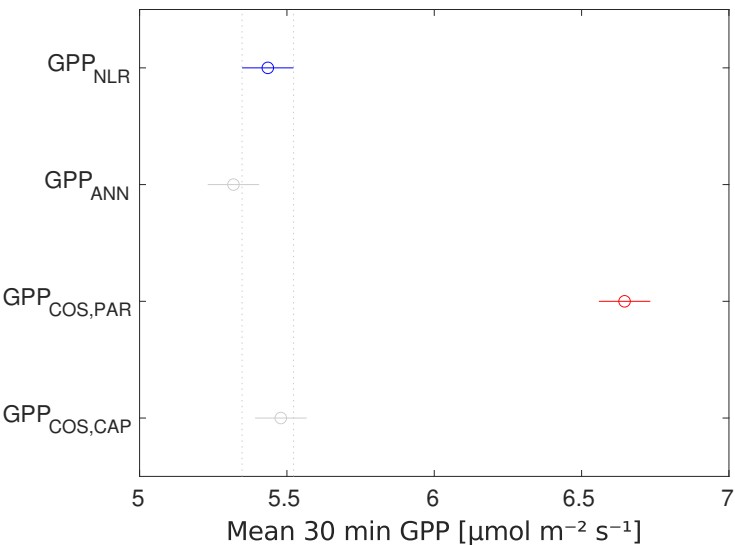

**Figure B4.** ANOVA test results for 30 min GPP data. Gray bars indicate no difference to the reference (blue) and red bars indicate statistical difference to the reference. The results show that only $GPP_{COS,PAR}$ differs statistically from $GPP_{NLR}$ on 30 min time scale.

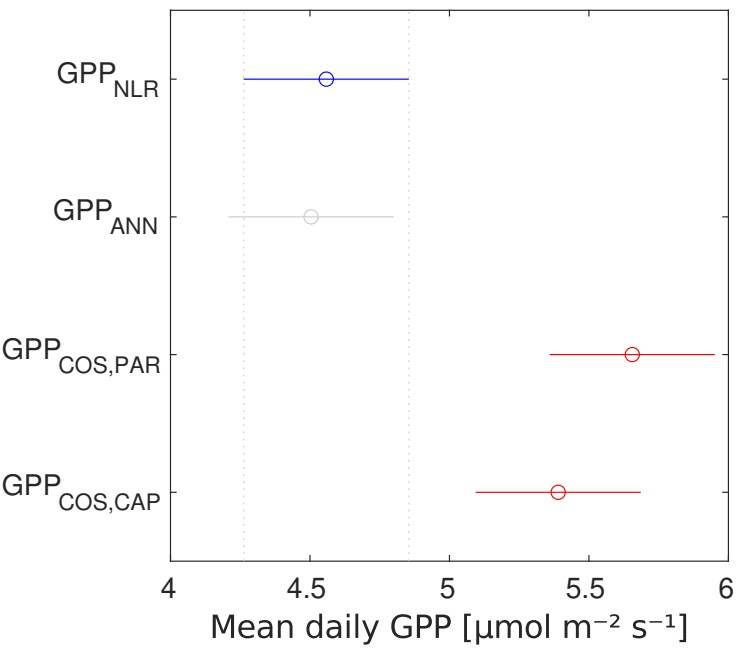

**Figure B5.** ANOVA test results for daily GPP data. Gray bars indicate no difference to the reference (blue) and red bars indicate statistical difference to the reference. The results show that both GPP$_{\text{COS,PAR}}$ and GPP$_{\text{COS,CAP}}$ differ statistically from both GPP$_{\text{NLR}}$ and GPP$_{\text{ANN}}$ on daily scale. GPP$_{\text{COS,PAR}}$ and GPP$_{\text{COS,CAP}}$ do not differ from each other.

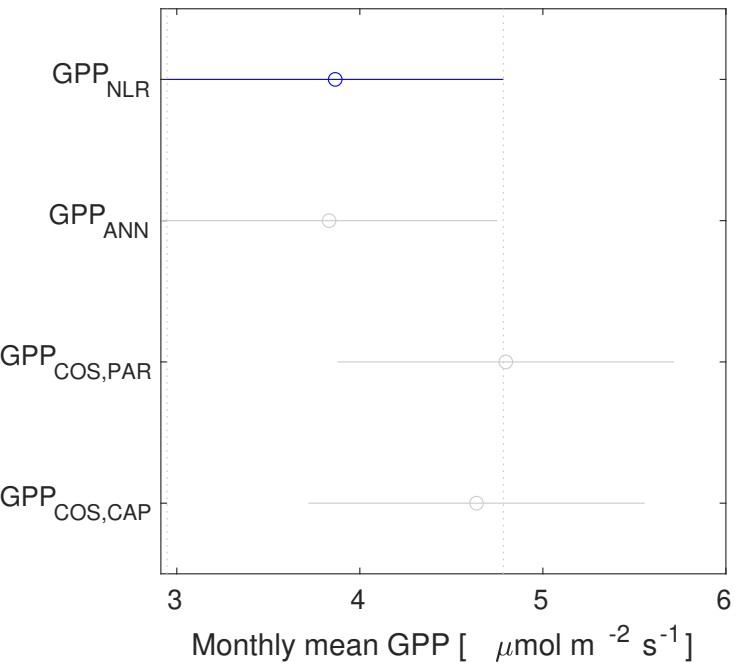

**Figure B6.** ANOVA test results for monthly GPP data. Gray bars indicate no difference to the reference (blue) and red bars indicate statistical difference to the reference. The results show that all GPPs are statistically the same on monthly scale.

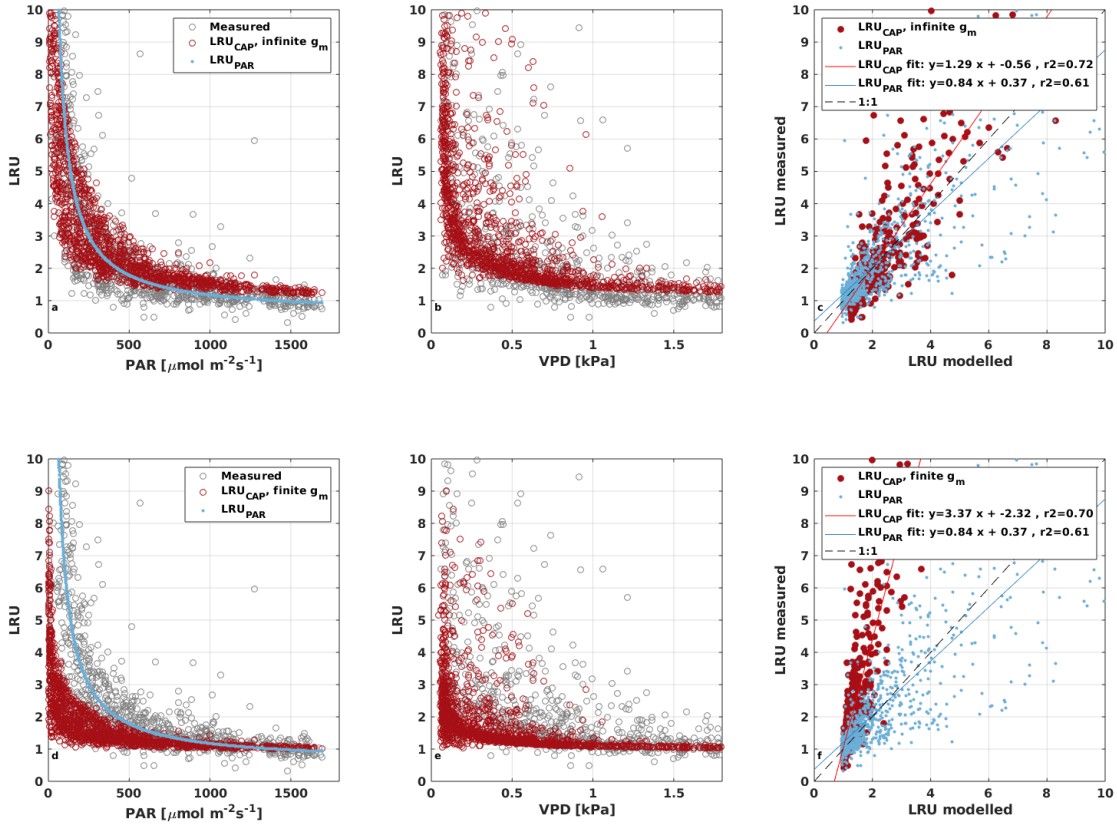

**Figure B7.** LRU derived from chamber measurements (gray) and modelled LRU$_{PAR}$ (blue) and LRU$_{CAP}$ (red) assuming infinite (a-c) or finite (d-f) mesophyll conductance ($g_m$) in LRU$_{CAP}$ against PAR and VPD. Subplots c and d compare the chamber measured LRU against modelled LRU$_{PAR}$ and LRU$_{CAP}$.

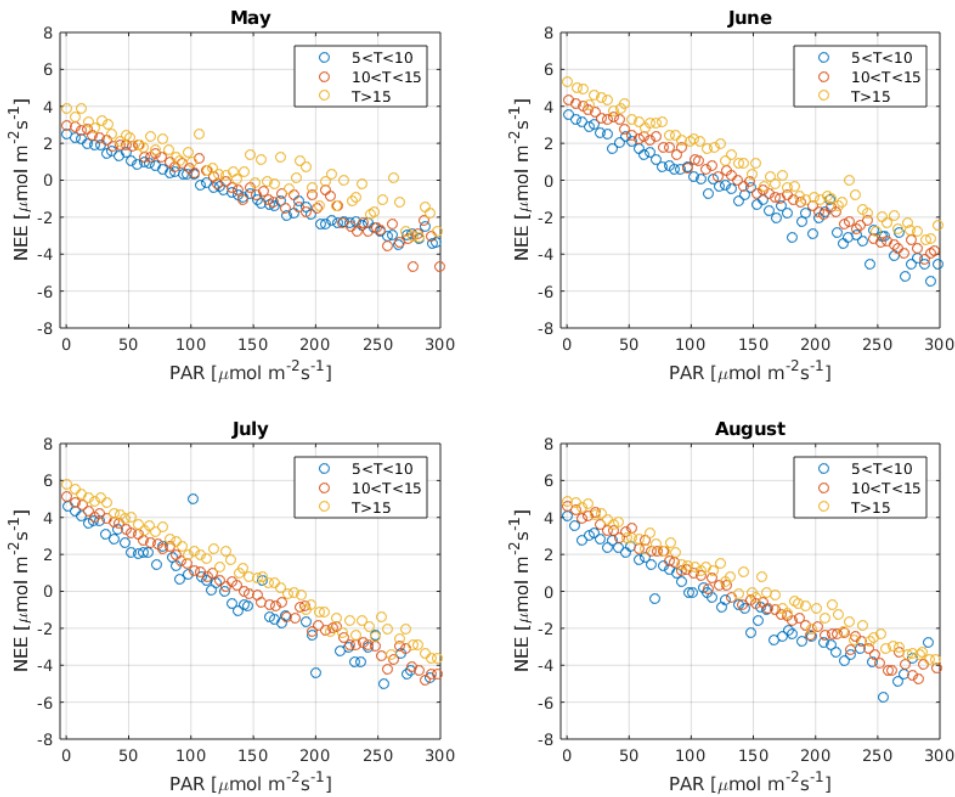

**Figure B8.** Net ecosystem exchange (NEE) against photosynthetically active radiation (PAR) close to the compensation point during May, June, July and August. Data are binned to different air temperature classes: $5°C < T_a < 10°C$ (blue), $10°C < T_a < 15°C$ (orange) and $T_a > 15°C$ (yellow).

*Author contributions.* KMK, IM and TV designed the study. KMK, PK and LMJK performed the measurements and flux processing. RD, AM and KMK developed the new LRU formulation. GT provided the GPP estimate by artificial neural networks. All authors contributed by commenting the study design, results and the manuscript. KMK wrote the manuscript with contributions from all co-authors.

*Competing interests.* The authors declare that they have no conflict of interest.

*Acknowledgements.* Special thanks to Helmi Keskinen, Sirpa Rantanen, Janne Levula, and other Hyytiälä technical staff for all their support with the measurements. The authors thank the Academy of Finland Centre of Excellence (118780), Academy Professor projects (312571 and 282842), ICOS Finland (3119871), ACCC Flagship funded by the Academy of Finland grant number 337549, Academy of Finland project 342930 and the Tyumen region government in accordance with the Program of the World-Class West Siberian Interregional Scientific and Educational Center (National Project "Nauka"). KMK thanks the Vilho, Yrjö and Kalle Väisälä foundation for its support. GT would like to acknowledge the support from the European Research Council (ERC) under the ERC-2017-STG SENTIFLEX project (grant agreement 755617). LMJK was supported by the ERC advanced funding scheme (AdG 2016, project no. 742798, project abbreviation COS-OCS). DP thanks the E-SHAPE H2020 project (grant agreement 820852) and the ICOS-ETC.

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
