# Peer review of "Intercomparison of methods to estimate gross primary production based on CO2 and COS flux measurements"

_Biogeosciences, 2022_

## Author Comment (AC1)

Reviewer comments in black
Author response in blue
*Changes in the revised manuscript in italic*

**Reviewer #1**

In the manuscript "Intercomparison of methods to estimate GPP based on CO2 and COS flux measurements", the authors compare 4 different models for calculating GPP on ecosystem level. Two of them are based on CO2 and environmental measurements, whereas the other 2 include a dependence on carbonyl sulfide fluxes. The GPP, based on a neural network, agreed very well with classic flux partitioning. The GPP based on the LRU of chamber measurements from the top of the canopy also agreed well with the classic approach, but tended to overestimate GPP during periods of high incoming photosynthetic radiation. The second COS based approach, using a stomatal optimization model, agreed much better with classic flux partitioning and, although its implementation to other field sites might be promising, still needs to be tested.

I generally agree, that this manuscript deserves to be published, but I have some questions and suggestions to improve the document.

We thank the reviewer for their insightful and helpful suggestions for improving the manuscript.

**General comments:**
- I suggest using an ANOVA and post-hoc tests to compare the results of 4 different models, daytimes and timescales instead of doing t-tests between only 2 of them. This could also end up in a nice table/plot for the reader. It's sometimes hard to grasp the differences within the text, which model results in a higher/lower GPP at different timescales and daytimes.

We thank the reviewer for this suggestion! We have performed ANOVA and post-hoc tests for the data set and will replace the t-test results with these. The result figures will be put in the appendix. The ANOVA test shows that $GPP_{ANN}$ and $GPP_{NLR}$ are not statistically different at any time scale, while $GPP_{COS,PAR}$ differs from $GPP_{NLR}$ at 30 min and daily time scales and $GPP_{COS,CAP}$ differs from $GPP_{NLR}$ only at daily time scale. $GPP_{COS,PAR}$ and $GPP_{COS,CAP}$ differ statistically only at 30 min time scale.

[Figure]

Figure 1: ANOVA test results for 30 min GPP data. Gray bars indicate no difference to the reference (blue) and red bars indicate statistical difference to the reference. The results show that only $GPP_{COS,PAR}$ differs statistically from $GPP_{NLR}$ at 30 min time scale.

[Figure]

Figure 2: ANOVA test results for daily GPP data. Gray bars indicate no difference to the reference (blue) and red bars indicate statistical difference to the reference. The results show that both $GPP_{COS,PAR}$ and $GPP_{COS,CAP}$ differ statistically from both $GPP_{NLR}$ and $GPP_{ANN}$ at daily scale. $GPP_{COS,PAR}$ and $GPP_{COS,CAP}$ do not differ from each other.

[Figure]

Figure 3: ANOVA test results for monthly GPP data. Gray bars indicate no difference to the reference (blue) and red bars indicate statistical difference to the reference. The results show that all GPPs are statistically the same at monthly scale.

- (Also, if a pairwise t-test is used to compare so many samples, the p-values need to be adjusted - see Bonferroni Holm).

We thank the reviewer for pointing this out. The t-test results will be replaced by the ANOVA test.

- I also think the publication would profit if you put plots showing the modeled versus the measured daytime NEE (for all approaches) for interested readers into the supplement (to compare over/underestimation of the models).

Thank you for this suggestion. While neither of the methods were developed in this paper, we will add comparison plots (see below) to the appendix for the interested reader. However, this comparison was limited to the NLR and ANN methods only, since we do not have an independent respiration estimate from COS fluxes in order to derive a model for NEE.

[Figure]

Figure 4: Modeled against measured NEE using (a) NLR and (b) ANN models for modeling NEE.

- Where do the differences between the daily and monthly GPPs averages come from?

Because of reduced noise when averaging over longer time. The differences, however, are relatively small.

- I suggest having an English native speaker proofread the manuscript since some sentences feel off.

The revised manuscript will be proofread by co-author Roderick Dewar, a native English speaker.

**Specific comments:**

38-40 The daytime approach of Lasslop et al. does not assume, that the respiratory processes are the same during day and nighttime, at least the base respiration is based on daytime data!

In Lasslop et al. (2010) they acquire the respiration related parameters from both nighttime and daytime data. When extrapolating their daytime model for daily and annual NEE, they write "For estimates of daily or annual NEE, respiration was extrapolated into the nighttime using $T_{air}$ measured during the night and the values obtained for $E_0$ and $r_b$." so in this sentence they do assume the repiratory processes to be the same during day and night, as the parameter values are assumed to be the same.

60 The carbonic anhydrase is also located within the cytoplasm. I would add this information. (see: Polishchuk, O. V. (2021). "Stress-Related Changes in the Expression and Activity of Plant Carbonic Anhydrases.")

We will add this information in the text: *"COS has been proposed as a proxy for GPP because it is taken up by plants through the same diffusive pathway as $CO_2$ and transported to the chloroplast surface. There it is destroyed by a hydrolysis reaction catalyzed by the enzyme carbonic anhydrase (CA, also located within the cytoplasm (Polishchuk, 2021)), while $CO_2$ continues its journey inside the chloroplast, where it is assimilated in the Calvin cycle (Wohlfahrt et al., 2012)."*

99 Was the friction velocity threshold applied during both day and nighttime?

Yes, as stated in the manuscript "*a threshold of 0.3 m s⁻¹ was applied to the whole data set*"

105 For the sake of completeness, it would also be nice to have the company/origin of pt100 sensor stated here.

Unfortunately, the company/origin of the sensor is unknown.

112 Why did you use 50% as a threshold?

We wanted to avoid days when most of the data was missing. Since COS measurements have a low signal-to-noise ratio, 50% was a good compromise to still include enough data, as e.g 75% would already be quite strict. In Vesala et al., (2022) 52 % of the COS flux data (the same data set that was used in this study) were discarded due to quality filtering in total. While this is a subjective tradeoff, it ensures that the analyzed daily GPPs originate as much as possible from measured fluxes and not too much from the gapfilling/partitioning procedure.We will clarify this in the revised manuscript as "*In Vesala et al., (2022), COS fluxes were found to have 52% data availability on average. While setting a 50% threshold is somewhat subjective, it ensures that the analyzed daily estimates of GPPs reflect measured fluxes rather than the gap-filling procedure.*"

168 I think the 4 th method should get a separate section including a title like the other 3.

We agree. We will separate the two LRU approaches into their own sections in the revised MS, as suggested.

186 Why did you use these exact values? Is there a reference for them? In which way will this influence the resulting LRU during the season regarding over and underestimation. It would be nice to see a sentence or two about this.

These values are representative of Scots pine at the Hyytiälä site. This will be clarified in the revised MS as: "*While Γ\* and α vary seasonally with temperature, for simplicity we used fixed values representing the growing season averages 50×10−6 mol mol⁻¹ and 0.05 mol mol⁻¹, respectively, (Bernacchi et al., 2001; Leverenz and Öquist, 1987).*"

Their influence on the resulting LRU is already revealed by our comparison of LRU obtained from the literature values of Γ\* and α with LRU obtained from best-fit values of parameters X and Y (in which Y depends on the ratio Γ\*/ α). Even though literature-based and fitted values of Y differ by a factor of three, the difference in LRU is relatively small (median difference of 4%). Although we explain this on L233-240, we will further clarify this point in the revised paper: "*This mismatch suggests there may be scope for further model improvement, such as the inclusion of dark respiration and/or finite mesophyll conductance in the LRU_{CAP} model. However, as the difference between fitted LRU_{CAP} and LRU_{CAP} with only literature values (statistical significance tested with Student's t-test, p<0.01) was not large with a median difference of 4 %, and the applicability of the model without fitting is better, we decided to use the literature value-based LRU_{CAP} in this study, without fitting to LRU_{ch}.*"

193 I am not sure the data is presented in a form, that will help the reader understand the data. Maybe it would be better to sort the sentences chronologically from 2013 to 2017 or group them by the environmental variables following Fig 1.

We will organize the text chronologically from 2013 to 2017 in the revised manuscript.

202 What does slightly higher mean? Can you give a percentage or absolute values?

The midday fluxes differed by 12 %. This information will be added in the revised manuscript: "*GPP_{ANN} showed on average 12 % higher midday values than GPP_{NLR} during summer months (May–July) in 2014 and 2017*"

209 Was the half-hourly data also statistically different?

ANOVA test for 30min fluxes show that $GPP_{NLR}$, $GPP_{ANN}$ and $GPP_{COS,CAP}$ do not differ statistically from each other, but $GPP_{COS,PAR}$ differed from all other methods. We will add discussion about this in the revised MS as: *"During the measurement period 2013–2017, 30 min, daily and monthly $GPP_{ANN}$ did not differ from $GPP_{NLR}$ statistically (tested with the ANOVA test; Fig. B3-B5).".*

210 How can GPPnlr be negative? Shouldn't the equation make it positive in any case, and set it to 0 when the air temperature is below 0?

$GPP_{NLR}$ is defined as GPP=R(modeled)-NEE(measured) whenever NEE measurements are available. When NEE measurements are missing, GPP is modelled with Eq. 2. This was not explained well enough in the previous version of the manuscript and we will add clarification and reorganize the text in the revised version. As there is always noise and uncertainty related to measured NEE, the GPP derived from flux measurements can also be negative if |NEE|>R.

215-216 The GPP difference in 4d is not for daily but for 30 min data, "e" shows daily values.

Thank you for noticing this, we will fix it in the revised MS.

232 LRUcap might be higher than LRU from chamber measurements. The LRU chamber measurements might not be best in representing the whole canopy. The higher LRU of GPPcap might even be closer to the true LRU value of the canopy, depending on the position of the chamber measurements. A higher LRU indicates more COS uptake per CO2 uptake, which might happen in the lower part of the canopy since there is less PAR, but the COS uptake should continue unhindered. I would refrain from concluding that LRUcap was overestimated during times of high radiation, but discuss the difference (and possible reasons) between the two COS based GPP.

This is a good point, the LRU from chamber measurements represents the top canopy only, therefore well-radiated conditions. $LRU_{CAP}$ is also calculated using PAR at the top of the canopy. Therefore differences between $LRU_{CAP}$ and LRU from chamber measurements may reflect limitations of the theory underlying $LRU_{CAP}$ (e.g. neglect of leaf respiration) rather than differences in canopy position. Differences between $GPP_{COS,PAR}$ and $GPP_{COS,CAP}$ most likely reflect intrinsic differences in the dependence of $LRU_{PAR}$ and $LRU_{CAP}$ on environmental drivers (PAR, VPD, SWC). This will be clarified in the revised MS as: *"However, it was noted that $LRU_{CAP}$ was higher than $LRU_{ch}$ and $LRU_{PAR}$ at high radiation (PAR > 1000 µmol $m^{-2}s^{-1}$, Fig. B6a). This may reflect intrinsic differences in the dependence of $LRU_{PAR}$ and $LRU_{CAP}$ on environmental drivers (PAR, VPD, SWC), as both of them represent LRU at the top of the canopy."*

234 I feel like introducing 2 new "parameters" in the result section is the wrong place, introduce them in the methods section.

The text introducing parameters X and Y will be moved to the Methods section 2.3.3 in the revised manuscript, as suggested.

239 Instead of writing "not large" can you tell if they are statistically different, and which one was higher/lower.

The RMSE of the modeled LRU to measured LRU decreased from 2.01 to 1.89 when performing the fitting (as mentioned on line 236). The fitted LRU was larger with a median value of 2.53 while without fitting 2.42, and the difference between these two methods was 4 %. They were also statistically different (tested with Student's t-test, p<0.01). This information will be added to the revised MS: *"However, as the difference between fitted $LRU_{CAP}$ and $LRU_{CAP}$ with only literature values (statistical significance tested with Student's t-test, p<0.01) was not large with a median difference of 4 %, and the applicability of the model without fitting is better, we decided to use the literature value-based $LRU_{CAP}$ in this study, without fitting to measured LRU. ".*

250-253 I don't think the comparison to a full season is needed, only state, that these cumulative measurements account for 13 weeks around the peak growing season.
Comparison to full growing season removed.

274 How did you find the saturation point. Which algorithm did you use?
We checked when diff(GPP)/diff(PAR) was less than 0.01. This happened at PAR>479 µmol $m^{-2}$ $s^{-1}$.

279 You could put a reference for Figure B2 here, showing the higher LRU at higher PAR
for LRUcap
Reference for Fig B2 will be added.

283 Usually, an increase in VPD should decrease the stomatal conductance/GPP. (see page 480-481 Körner, C. (1995). Leaf Diffusive Conductances in the Major Vegetation Types of the Globe. Ecophysiology of Photosynthesis. E.-D. Schulze and M. M. Caldwell. Berlin, Heidelberg, Springer Berlin Heidelberg: 463-490. and Lasslop, G., et al. (2010). "Separation of net ecosystem exchange into assimilation and respiration using a light response curve approach: critical issues and global evaluation." Global Change Biology 16(1): 187-208. The correlation between VPD and GPP in spring might only be caused by the correlation of the air temperature with VPD.
We will add clarification in the revised MS: *"However, the apparent increase in GPP with VPD in spring could be caused by the correlation of $T_a$ with VPD, coinciding with the start of the growing season, as the trees are not water-limited after snow melt."*

286-289 Since you have not observed a drought or heatwave, these sentences feel unnecessary.
The sentences about drought/ heatwave will be removed from the revised MS.

303 State the difference here, "similar" feels unclear.
The revised manuscript will state *"...we observed a 25% difference in the midday GPP during summer, similar to what was found in Kooijmans et al., (2019)..."*

316 I actually dislike the term measured LRU, as LRU is a product of GPP and COS fluxes, so it can't be measured. I suggest replacing measured LRU with "LRU derived from chamber measurements" or a wording that is more representative for the LRUs calculation.
We agree with the reviewer and will change "measured LRU to "LRU derived from chamber measurements" or shorthand $LRU_{ch}$ throughout the revised manuscript.

318 Why was it not comparable to the chamber measured LRU?
With this sentence we meant that the finite gm method didn't compare as well as the infinite gm method (agreement was not as good) with the LRU derived from chamber measurements. We will clarify this sentence in the revised MS and add quantification of the disagreement: *"We also provide a formulation of $LRU_{CAP}$ with finite $g_m$, which did not compare well with $LRU_{ch}$ at Hyytiälä forest (RMSE=2.58, median difference to $LRU_{ch}$ 22%), especially during low light conditions, but could compare better at other measurement sites".*

322 It would be nice to have the information about the position of the leaf chambers in the methods section, so that the reader knows what the basis for the LRUpar is.
We will add this information in the methods section 2.3.3 as: *"This LRU equation was based on field measurements of pine branch $CO_2$ and COS fluxes with two chambers placed at the top of the canopy in 2017 at the same site and were thus independent from the EC flux measurements (Kooijmans et al., 2019)."*

336 I am not sure that you should conclude that the LRUcap model underestimates GPP during midday compared to GPPpar. Due to the aforementioned issue of only having a chamber at the top

of the canopy, the GPPpar might be overestimated and GPPcap could actually better. (The LRU at the top of the canopy might be lower compared to areas within the canopy). I propose, just writing that the GPP is lower instead of underestimated, since "underestimated" gives the impression that GPPpar is correct.

*We have changed the wording "underestimated" to "lower", as suggested.*

Fig 3 Is this figure based on half-hourly data points? Are these really average or median differences like in Fig 2?

*It is the median difference between the methods, i.e. the differences to GPP$_{NLR}$ in Fig. 2*

Fig 5 Do you mean cumulative daily fluxes when you write daily flux data points? You mention, that all medians have been calculated using the same number of data points. Were these also the same data points, or could there be a bias from different days?

*This figure presents the differences in the daily median fluxes (not cumulative). Also here the daily medians have been calculated with only the same exact data points.*

Fig 7 Why did you use 700 par as the threshold?

*At PAR>700 µmol m$^{-2}$ s$^{-1}$ the GPP vs PAR curve starts to saturate so there is no more radiation dependence interfering with the intercorrelated temperature and VPD.*

Fig B2 If measured means "chamber -measured" LRU please state so.

*Yes, corrected as suggested.*

**Technical corrections:**
91 "Consisted of a Gill HS"

*Corrected as suggested.*

112 I feel like some words are missing in this sentence. The second part about monthly averages feels disconnected. Are you trying to say, that the monthly averages were also only calculated from daily means, when 50% of the half-hourly data was available?

*Yes, corrected in the revised MS as "Daily average GPP was only calculated if more than 50% of measured 30-min flux data was available for each day, and monthly averages were calculated from the daily means."*

218 To investigate further the causes for the …

*Corrected as suggested.*

253 remove brackets from (on average 25%)

*Corrected as suggested.*

332 Do you mean noisy (scattered)?

*Corrected as suggested.*

**Reviewer #2**
The study by Kohonen et al. compares gross primary productivity (GPP) estimates at a boreal forest derived from two CO2 -based flux partitioning methods and two COS-based methods. One of the COS approaches to GPP, developed in previous studies, relies on an empirical light response of the COS vs CO2 leaf relative uptake (LRU) ratio. The other COS approach, developed in this study, considers stomatal optimization as represented by the CAP model (Dewar et al., 2018) in simulating

LRU responses to environmental conditions. The authors show that GPP estimates derived from the LRU CAP approach agree with those from the two CO2 -based approaches in terms of diurnal and seasonal cycles, cumulative GPP in the growing season, and environmental responses. By contrast, the COS approach based on the light dependence of LRU alone shows considerably higher GPP estimates than those from other methods, especially at high radiation. The authors conclude that their new approach is an improvement over previous empirical LRU fits for obtaining accurate COS-based GPP estimates.

Overall, the study marks a valuable methodological advance in estimating GPP at the ecosystem scale and is worthy of publication. While the authors succeed in deriving COS-based GPP estimates consistent with those from CO2 -based methods, they have not presented a strong case for the robustness and generalizability of the new method they developed. In other words, do we know that the LRU CAP approach produces the right results for the right reason, or is it so malleable that one can tune the parameters to get any desirable responses? To ensure the robustness of the method, the authors may need to clarify the physiological underpinnings of the method, the assumptions it makes, and its limitations. I have a few questions on this aspect.

We thank the reviewer for the insightful and helpful comments to improve the manuscript.

In the revised paper, and especially in the Appendix, we explain more clearly that $LRU_{CAP}$ is based on a generic physiological model of stomatal function whose robustness has been established previously (e.g. Lintunen et al. 2019; Salmon et al. 2020; Dewar et al. 2021, Gimeno et al., 2019). The model parameters are all physiologically meaningful, and can be measured independently or obtained from literature. No parameter tuning is required. This represents a clear advance on previous COS-based methods based on empirical fitting ($LRU_{PAR}$). It is true that in our study, in order to gauge the sensitivity of LRU to the model parameters, we compared $LRU_{CAP}$ calculated from literature-based parameters to $LRU_{CAP}$ obtained by fitting the parameters X and Y, but this is not a necessary requirement for applying $LRU_{CAP}$. This will be made more clear in the revised MS, both in the methods and Appendix sections. The Appendix will be revised thoroughly and a paragraph added to methods "*$LRU_{CAP}$ is based on a generic physiological model of stomatal function whose predictions have been successfully tested previously (e.g. Lintunen et al. (2020); Salmon et al. (2020); Dewar et al. (2021); Gimeno et al. (2019)). The model parameters are all physiologically meaningful, and can be measured independently or obtained from the literature. This formulation therefore represents a clear advance on previous COS-based methods based on empirical fitting ($LRU_{PAR}$), because it provides a physiological explanation for variations in LRU that may be more robust when extrapolating to other sites.*"

- There are many optimization-based stomatal models, and CAP is not the simplest one. What is the motivation for choosing this specific model over, say, the Medlyn model (Medlyn et al., 2011), which has only two parameters to fit?

  As noted above, the advantage of using CAP is that, unlike the Medlyn et al model, it has no undetermined parameters and therefore does not require parameter fitting. In addition, CAP takes into account soil-to-leaf hydraulics, which makes it theoretically more ambitious than most of the simpler models. The advantages of CAP will be clarified in the revised manuscript methods, as described above.

- The "carboxylation conductance", gc, seems to be a pure model construct to linearize the nonlinear response of the assimilation rate (A) to the chloroplast CO2 concentration ($c_c$). The assumption that gc is constant is inconsistent with the Farquhar et al. (1980) model because the transition from Rubisco carboxylation limitation to electron transport limitation necessarily changes the slope of the A–$c_c$ curve. What is the rationale behind this treatment? What bias does it introduce?

  This is not correct. The A-$c_c$ response underlying CAP is non-linear and is derived from a simplified representation of the light and dark reactions of photosynthesis (Thornley &

Johnson 1990). Parameter $g_c$ is the initial slope of this non-linear response, and is equivalent to the parameter combination Vcmax/km of the Farquhar model. The rationale for using the Thornley-Johnson photosynthesis model is that in the Farquhar model the abrupt switch from Rubisco- to electron transport limitation introduces artificial discontinuities in the solution for optimal stomatal conductance, whereas in the T-J model there is a smooth transition from CO2- to light limitation and no such discontinuities occur. No bias is introduced. The revised paper will make these points more clearly (especially the Appendix).

- Several parameters assumed constant in fitting the model may vary across the season, for example, CO2 compensation point and photosynthetic quantum yield. Where do those fixed values come from? Are they representative of the Scots pine species at the site?
  These values are representative of Scots pine at the Hyytiälä site. This has been clarified in the revised MS as *"While Γ* and α vary seasonally with temperature, we decided to use fixed values representing the growing season averages 50×10−6 mol mol⁻¹ and 0.05 mol mol⁻¹, respectively, for simplicity (Bernacchi et al., 2001; Leverenz and Öquist, 1987).".* Their influence on the resulting LRU is already revealed by our comparison of LRU obtained from the literature values of Γ* and α with LRU obtained from best-fit values of parameters X and Y (in which Y depends on the ratio Γ*/ α). Even though literature-based and fitted values of Y differ by a factor of three, the difference in LRU is relatively small (median difference of 4 %).  Although we explain this on L233-240, we will further clarify this point in the revised paper as *"This mismatch suggests there may be scope for further model improvement, such as the inclusion of dark respiration and/or finite mesophyll conductance in the LRU$_{CAP}$ model. However, as the difference between fitted LRU$_{CAP}$ and LRU$_{CAP}$ with only literature values (statistical significance tested with Student's t-test, p<0.01) was not large with a median difference of 4 %, and the applicability of the model without fitting is better, we decided to use the literature value-based LRU$_{CAP}$ in this study, without fitting to LRU$_{ch}$.".*

- The impact of mesophyll conductance (gm) on LRU is an intriguing but understated point. It seems that infinite gm works best for explaining LRU variability at low light but overestimates LRU at high light. By contrast, a finite gm works well at high light but predicts too low LRU values at low light (Fig. B2). Is there a physiological explanation for this? A discussion on this point would be desirable.
  The general expression for LRU given by Eqn (7), which is based on flux balance alone (i.e. independent of assumptions about stomatal behaviour), shows that LRU is higher for infinite gm than for finite gm. This is indeed the case for our predictions of LRU$_{CAP}$ with infinite  vs. finite gm (cf. Eqns A2 and A11 with $c_c < c_i$ when gm is finite; this is also apparent from Fig. B2). In CAP, these two cases represent two contrasting hypotheses, in which non-stomatal limitations (NSLs) act either entirely on photosynthetic capacity, or entirely on gm, respectively. In reality, NSLs may act on both photosynthetic capacity and gm, with one or other effect being dominant depending on environmental conditions. The contrasting abilities of each hypothesis to explain chamber-measured LRU at low vs. high light, as noted by this reviewer, might be explained by a shift in the action of NSLs from photosynthetic capacity to gm as light increases. However, verifying this possibility lies beyond the scope of the present study. The revised paper will discuss these points: *"We find a better agreement of LRU$_{CAP}$ with LRU$_{ch}$ if $g_m$ is assumed infinite, but there is a mismatch at high PAR, supporting the possibility that gm might indeed be a limiting factor under high radiation. In CAP, infinite or finite gm represent two contrasting hypotheses, in which NSLs act either entirely on photosynthetic capacity, or entirely on $g_m$, respectively. In reality, NSLs may act on both photosynthetic capacity and $g_m$, with one or other effect being dominant depending on environmental conditions. The contrasting abilities of each hypothesis to*

*explain LRUch at low vs. high light, might be explained by a shift in the action of NSLs from photosynthetic capacity to gm as light increases. However, verifying this possibility lies beyond the scope of the present study."*

**Specific comments**

L21–22: "removes approximately 30% of the annual anthropogenic carbon dioxide (CO2) emissions from the atmosphere". This is a misinterpretation. Global GPP far outweighs the anthropogenic carbon emissions (~120 PgC vs ~10 PgC). The 30% fraction refers to net biome productivity, which is the net balance of GPP, ecosystem respiration, and emissions from land use changes and disturbances. See Chapin et al. (2006) for standard definitions of carbon flux terms.
We thank the reviewer for pointing this out. We will modify the sentence to *"Photosynthetic carbon uptake (or gross primary production, GPP) is a key component of the global carbon cycle, with the terrestrial ecosystems removing approximately 30 % of annual anthropogenic carbon dioxide (CO₂) emissions from the atmosphere"* to avoid misconceptions.

L25: It is the net balance not the ratio that dictates the magnitude and direction of the terrestrial carbon budget.
Changed the wording from "ratio" to "rate".

L33: The origin of the partitioning method based on nighttime respiration predates Reichstein et al. (2005). The idea goes back at least as early as in Wofsy et al. (1993), though not in the exact form of relationship between Reco and temperature. It is likely that this method has an earlier origin in the eddy covariance community. Therefore, better change "a method introduced by Reichstein et al. (2005)" to "a method in Reichstein et al. (2005)".
Corrected as suggested.

L35: And storage change fluxes, if not constrained by concentration profile measurements, also introduce bias to nighttime fluxes.
As neglecting storage change adds bias to the whole daily cycle instead of nighttime only, we have not specified this in this sentence (that focuses on nighttime problems only), as it is not so relevant in the introduction given the scope of our manuscript.

L40: "These limitations lead to uncertainties in the derivation of mechanistically sound descriptions of respiration and its drivers, especially when contributions of different biomass compartments to total CO2 efflux vary across ecosystems and seasonally even within one ecosystem." The point of this sentence is unclear.
The sentence will be modified as: *"These assumptions lead to uncertainties in partitioning because different biomass compartments (soil organic matter, roots, stems, branches, foliage) could have different drivers and respiration responses even within the same ecosystem"*

L48–55: It would be helpful to add a sentence on how this neural network approach tackles the problem of the inhibition of daytime respiration.
NNC-part is a partitioning method based on machine learning, hence it is data-driven. However, the network structure emulates the light use efficiency concept and thus gross photosynthesis is partially constrained. Instead, the Kok effect is not explicitly accounted for and Tramontana et al., 2020, reports: "…it is not possible to demonstrate that the NNC-part method, as implemented in this experiment, is able to reproduce the light inhibition of leaf respiration."
In the revised version of the manuscript, the physiological value aspects of NNC-part will be clarified by adding the following sentence in L141. *"NNC-part has a hybrid nature and gross photosynthesis is partially constrained by emulating the LUE concept."*

L66: "recent studies have shown that LRU is a function of solar radiation because CO2 uptake is highly radiation dependent while COS uptake is not" - This notion that LRU depends on PAR goes back as early as Stimler et al. (2010).

Added reference to Stimler et al. (2010).

L123: Specify the value of T0.

$T_0$=-2°C, specified in the revised manuscript.

Section 2.3.2: Did you create a hold-out data set for validation as in Tramontana et al. (2020), or perform cross-validation?

The artificial neural network processing scheme is the same as in Tramontana et al., 2020, except for some details that we have clarified in the method section of the current manuscript version (please see section 2.32, ln 146-140). These changes did not affect model validation.

L161: "atmospheric concentrations of CO2 and COS" - Specify at which height these concentrations were measured.

Specified in the revised MS as "*at the EC measurement height*"

L164: Kooijmans et al. (2019) presented data from two chambers. Was this relationship derived from measurements from both chambers?

Yes, we use the average of the two chambers, like in Kooijmans et al. (2019). Specified in the revised MS as "*This LRU equation was based on field measurements of branch $CO_2$ and COS fluxes with two chambers in 2017 at the same site and were thus independent from the EC flux measurements (Kooijmans et al., 2019).*"

L193–200: I share the other referee's concern that this paragraph is not helpful for readers to grasp the year-to-year variability of environmental conditions. Try to present the anomaly features in chronological order.

Text will be organized chronologically in the revised manuscript as suggested.

Table 1: List the source of each parameter value in a column instead of in the caption. Specify which values are from the literature and which are fitted to data presented in this study.

The sources of each parameter will be added in the table in the revised MS, as suggested.

L203–204: "... when comparing GPP ANN to standard FLUXNET partitioning during summer months for multiple sites." - What about the subset of evergreen needleleaf forest (ENF) sites?

In order to answer the reviewer's question there are two plots below derived from data produced used in Tramontana et al., 2020 that show the mean diurnal cycle of GPP for a subset of Boreal ENF (Latitude > 50°N) and for Hyytiälä site. The systematic differences among $NN_{C-part}$ and standard partitioning methods seem very consistent with the patterns reported in Tramontana et al., 2020; the dynamics calculated for Hyytiälä study site seem consistent with this general trend and with the findings of our manuscript. However, it is important to remember that there are few differences between data used in Tramontana et al., 2020 and the one used in this manuscript concerning both NEE flux processing and NLR relationships applied as partitioning methods. For this reason a direct comparison between the two studies in not possible and lies beyond the scope of the submitted manuscript.

[Figure]

Figure 5: Differences of nighttime and daytime partitioning methods to NN_C-part in the evergreen needleleaf forests (ENF, latitude > 50°N).

[Figure]

Figure 6. Differences of nighttime and daytime partitioning methods to NN_C-part in Hyytiälä forest.

L209–L210: "However, at 30 min time scale the GPP ANN was on average 15 % lower than GPP NLR ." - Could you compare GPP ANN and GPP NLR at half-hourly timescales with negative values filtered?

GPP_NLR contains negative values due to measurement uncertainty in NEE measurements. These negative values often appear during nighttime or low-light periods. GPP_ANN is a "model" produced by machine learning and is not allowed to have negative values. Filtering out negative GPP_NLR values would skew its distribution and we want to keep the results as independent as possible from any interference from the data user.

L211–L212: "while GPP NLR may have even negative values due to random noise in the NEE measurements." - GPP should not be negative. Even if we consider random noise, the uncertainty range of GPP estimates should not encompass negative values because this is physically impossible. In your calculation of cumulative fluxes, the negative values may need to be capped at zero.
While it is true that by definition GPP should not be negative, the estimated process rate derived from flux measurements may have negative values due to measurement uncertainty and noise. However, on average the nighttime GPP is zero.
Forcing negative GPP to zero would skew its distribution at low light and bias the cumulative GPP.

L231: Given that GPP is higher at high radiation, shouldn't the parameter fitting prioritize reducing LRU bias at high radiation?
There were no parameters fit to the basic $LRU_{CAP}$ model (referring to line 231). However, we did try fitting parameters $X = |\psi_c| / 1.6g_c$ and $Y = 2\Gamma^*g_c / \alpha$ to $LRU_{CAP}$ instead of using literature values. This fitting was done against the measured LRU, not GPP. LRU values are low under high radiation and high under low radiation. Due to the logarithmic nature of LRU, the fitting was done to log(LRU), as explained in the appendix, Sect. A1.

L242: "The agreement of this method was better than assuming infinite mesophyll conductance at high PAR, but worse at low PAR" - Could you elaborate on why this is the case? Have you tried temperature-dependent gm as in Wehr et al. (2017)?
See response to general comment on this point above. The T-dependence of gm lies outside the scope of this study.

L245–246: "We thus concluded that the assumption of infinite gm is more valid." - It would be more appropriate to say that given the uncertainty in LRU, minimizing LRU errors by itself does not offer a robust constraint on gm . This fact does not necessarily mean that an infinite gm is valid in the real world.
This was badly phrased. In the revised paper we will rephrase this as "… *the assumption of infinite $g_m$ gives an estimate closest to the LRU derived from chamber measurements, although the assumption in itself is  physiologically unrealistic.*"

L249: It is worth noting that gm becomes more limiting relative to gs. We do not know how gm varies during the day. It could be that gs increases to a point such that gm becomes more limiting.
As we already stated above, our results are consistent with the action of NSLs changing from photosynthetic capacity to gm, but further analysis of this possibility lies beyond the scope of this study.

L267–L269: If the fraction of leaf respiration in total ecosystem respiration is small, I would not expect a clear break point to be found in the light response of NEE. Do you see any evidence for the Kok effect in leaf chamber measurements?
Studying the Kok-effect requires separation of photosynthesis and respiration at the leaf scale, which may require additional measurements other than $CO_2$, e.g. $^{13}C$ and $^{18}O$ in $CO_2$, and cannot be independently derived for the hourly measurements.  Daytime foliage respiration upscaled from chambers is typically 2-3 µmol m$^{-2}$ s$^{-1}$ in the growing season. So the magnitude of Kok effect would be just tenths of µmol, not very much compared to the uncertainties of (nighttime) eddy fluxes and extrapolating T response to daytime.  Therefore, we do not further discuss the Kok-effect in the manuscript.

L274: "in summer a saturation point was found at PAR>500" - This apparent saturation point could be partly caused by VPD limitation on stomatal conductance around midday.
True, we will add this note in the revised manuscript as "..., *that could be linked to VPD limitation on stomatal conductance in the afternoon (Kooijmans et al., 2019).*".

L359: What purpose does rewriting the equation in terms of $c_a - \Gamma *$ serve? In the Farquhar et al. (1980) model, $\Gamma *$ appears in $c_c - \Gamma *$, because it is used to represent the difference between carboxylation and oxygenation. But $c_a - \Gamma *$ does not seem to carry a physiological meaning.

The reason for rewriting LRU in terms of $c_a - \Gamma *$ is that CAP predicts a simple expression for the ratio $(c_i - \Gamma *)/(c_a - \Gamma *)$ whereas LRU is related more directly to $c_i/c_a$. In the original ms, this reason is not apparent at first, because of the order in which the equations are presented. In the revised paper we will rewrite the equation for LRU in terms of $(c_i - \Gamma *)/(c_a - \Gamma *)$ only after we have presented the CAP prediction for $(c_i - \Gamma *)/(c_a - \Gamma *)$, so that the rationale for doing so is clearer. The reason $(c_i - \Gamma *)/(c_a - \Gamma *)$ emerges from CAP, rather than $c_i/c_a$, is precisely the one the reviewer refers to: in the underlying photosynthesis model (with infinite $g_m$) the $CO_2$ dependence occurs through $c_i - \Gamma *$ (reflecting carboxylation minus oxygenation).

**Technical comments**

L24: "increased" -> "increasing"
Corrected as suggested.

L30: "widely" and "globally", superfluous
Removed "globally".

L61: "triggered" -> "catalyzed"
Corrected as suggested.

L69: "ecosystem scale" -> "ecosystem-scale"
Corrected as suggested.

L71–72: This sentence seems to be the topic sentence of the paragraph.
The sentence was moved to the beginning of the paragraph in the revised manuscript.

L84: "where first flux measurements started in 1996 ..." - This information does not seem relevant since only the flux measurements between 2013 and 2017 are presented.
Removed as suggested.

L86: "50 ha" - Better use SI units, for example, 0.5 km 2 .
Hectare is also an SI unit and most commonly used to describe forest area. We decided to leave as it was.

L139: "ecosystem level" -> "ecosystem-level"
Corrected as suggested.

L146: "assure" -> "ensure"
Corrected as suggested.

L193: "higher average" -> "higher than average"
Corrected as suggested.

L195: The units of PAR are incorrect in this line.
We thank the reviewer for noticing this! Corrected to $\mu mol\ m^{-2}\ s^{-1}$.

L214: "Fig. 2,3" -> "Figs. 2 and 3"
Corrected as suggested.

**References cited**

Chapin, F. S., Woodwell, G. M., Randerson, J. T., Rastetter, E. B., Lovett, G. M., Baldocchi, D. D., Clark, D. A., Harmon, M. E., Schimel, D. S., Valentini, R., Wirth, C., Aber, J. D., Cole, J. J., Goulden, M. L., Harden, J. W., Heimann, M., Howarth, R. W., Matson, P. A., McGuire, A. D., … Schulze, E.-D. (2006). Reconciling Carbon-cycle Concepts, Terminology, and Methods. Ecosystems, 9(7), 1041–1050. https://doi.org/10.1007/s10021-005-0105-7

Dewar, R., Mauranen, A., Mäkelä, A., Hölttä, T., Medlyn, B., & Vesala, T. (2018). New insights into the covariation of stomatal, mesophyll and hydraulic conductances from optimization models incorporating nonstomatal limitations to photosynthesis. New Phytologist, 217(2), 571–585. https://doi.org/10.1111/nph.14848

Farquhar, G. D., von Caemmerer, S., & Berry, J. A. (1980). A biochemical model of photosynthetic $CO_2$ assimilation in leaves of C3 species. Planta, 149(1), 78–90. https://doi.org/10.1007/BF00386231

Kooijmans, L. M. J., Sun, W., Aalto, J., Erkkilä, K.-M., Maseyk, K., Seibt, U., Vesala, T., Mammarella, I., & Chen, H. (2019). Influences of light and humidity on carbonyl sulfide-based estimates of photosynthesis. Proceedings of the National Academy of Sciences, 116(7), 2470–2475. https://doi.org/10.1073/pnas.1807600116

Medlyn, B. E., Duursma, R. A., Eamus, D., Ellsworth, D. S., Prentice, I. C., Barton, C. V. M., Crous, K. Y., De Angelis, P., Freeman, M., & Wingate, L. (2011). Reconciling the optimal and empirical approaches to modelling stomatal conductance. Global Change Biology, 17(6), 2134–2144. https://doi.org/10.1111/j.1365-2486.2010.02375.x

Reichstein, M., Falge, E., Baldocchi, D., Papale, D., Aubinet, M., Berbigier, P., Bernhofer, C., Buchmann, N., Gilmanov, T., Granier, A., Grunwald, T., Havrankova, K., Ilvesniemi, H., Janous, D., Knohl, A., Laurila, T., Lohila, A., Loustau, D., Matteucci, G., … Valentini, R. (2005). On the separation of net ecosystem exchange into assimilation and ecosystem respiration: Review and improved algorithm. Global Change Biology, 11(9), 1424–1439. https://doi.org/10.1111/j.1365-2486.2005.001002.x

Stimler, K., Montzka, S. A., Berry, J. A., Rudich, Y., & Yakir, D. (2010). Relationships between carbonyl sulfide (COS) and $CO_2$ during leaf gas exchange. New Phytologist, 186(4), 869–878. https://doi.org/10.1111/j.1469-8137.2010.03218.x

Tramontana, G., Migliavacca, M., Jung, M., Reichstein, M., Keenan, T. F.,Campsâ, Valls, G., Ogee, J., Verrelst, J., & Papale, D. (2020). Partitioning net carbon dioxide fluxes into photosynthesis and respiration using neural networks. Global Change Biology, 26(9), 5235–5253. https://doi.org/10.1111/gcb.15203

Wehr, R., Commane, R., Munger, J. W., McManus, J. B., Nelson, D. D., Zahniser, M. S., Saleska, S. R., & Wofsy, S. C. (2017). Dynamics of canopy stomatal conductance, transpiration, an evaporation in a temperate deciduous forest, validated by carbony sulfide uptake. Biogeosciences, 14(2), 389–401. https://doi.org/10.5194/bg-14-389-2017

Wofsy, S. C., Goulden, M. L., Munger, J. W., Fan, S.-M., Bakwin, P. S., Daube, B. C., Bassow, S. L., & Bazzaz, F. A. (1993). Net Exchange of CO 2 in a Mid-Latitude Forest. Science, 260(5112), 1314–1317. https://doi.org/10.1126/science.260.5112.1314

**References**

Dewar, R., Hölttä, T., & Salmon, Y. (2021). Exploring optimal stomatal control under alternative hypotheses for the regulation of plant sources and sinks. *New Phytologist, 233*(2), 639-654.

Gimeno, T. E., Saavedra, N., Ogée, J., Medlyn, B. E., & Wingate, L. (2019). A novel optimization approach incorporating non-stomatal limitations predicts stomatal behaviour in species from six plant functional types. Journal of experimental botany, 70(5), 1639-1651.

Lintunen, A., Paljakka, T., Salmon, Y., Dewar, R., Riikonen, A., & Hölttä, T. (2020). The influence of soil temperature and water content on belowground hydraulic conductance and leaf gas exchange in mature trees of three boreal species. Plant, Cell & Environment, 43(3), 532-547.

Salmon, Y., Lintunen, A., Dayet, A., Chan, T., Dewar, R., Vesala, T., & Hölttä, T. (2020). Leaf carbon and water status control stomatal and nonstomatal limitations of photosynthesis in trees. New phytologist, 226(3), 690-703.

Vesala, T., Kohonen, K. M., Kooijmans, L. M., Praplan, A. P., Foltýnová, L., Kolari, P., ... & Mammarella, I. (2022). Long-term fluxes of carbonyl sulfide and their seasonality and interannual variability in a boreal forest. Atmospheric Chemistry and Physics, 22(4), 2569-2584.

---

## Author Response (AR2)

Reviewer comments in black
Response in blue
*Text in the revised MS in italic*

**Reviewer #1**
This is the second time I review the manuscript by Kohonen et al. I believe the major points have been adequately addressed by the authors. The overall flow and clarity of the manuscript have been improved substantially. I have attached a few minor and technical comments below.
We thank the reviewer for the further comments and suggestions to improve the manuscript.

L14: I believe a comma is advisable before "leading to ...".
Corrected as suggested.

L17: "can be estimated from simple meteorological measurements or the literature" - And leaf and soil hydraulic parameters?
The parameters referred to in this sentence include all the model parameters in Table 1 (meteorological, environmental, leaf photosynthetic, xylem hydraulic, soil hydraulic, …). As this is the abstract, there is no room to go into any further detail about the specific nature of the parameters. Thus, we have simplified the sentence to "*can be estimated from simple measurements or obtained from the literature*".

L53: "proxies for CO2 uptake" -> "proxies for photosynthetic CO2 uptake"
Corrected as suggested

L121: "the mean of the air temperature at 18 m height" - On line 103, you mentioned that air temperature was measured at 16.8 m.
Thank you for noticing this, it was corrected to 16.8m, where the temperature measurements indeed took place.

L200, L251--252: Add units for X and Y.
Added units for X and Y: $[X] = MPa\ m^2\ s\ mol^{-1}$; $[Y] = mol\ m^{-2}\ s^{-1}$

L230--233: The absence of a statistically significant difference on the monthly scale feels perplexing (Figs. B4 and B5). Why do GPP_LRU and GPP_CAP differ substantially on 30-min and daily scales but not on the monthly scale? Is there any difference in monthly aggregation or gap-filling? It would be helpful to give a brief explanation as to why this is the case.
The monthly values are means from the daily values, which are calculated from gap-filled 30min flux data so that at least 50% of fluxes need to be measured each day (meaning at least 24 datapoints out of total 48). Averaging reduces scatter, and thus difference to the reference GPP. There is no difference in gap-filling between the time scales, as gap-filling is done to 30min fluxes only, from which the daily and monthly averages then yield. However, there is an increased uncertainty in the ANOVA test on monthly scale, as the sample size is reduced from 23994 (30min flux values) to 30 (monthly fluxes).

L239: "However, there is less scatter ..." - A reference to Fig. 4d,g may be included here.
Figure reference added.

L255: "as the difference between fitted LRU_CAP and literature-based LRU_CAP (statistical significance tested with Student's t-test, p<0.01) was not large ..." - I find this statement unintuitive. The parameter Y determines the light dependence of LRU, and if it is increased by a factor of 3.3, the term sqrt(1 + Y / PAR) would increase by a factor of 1.8 when PAR = 10, or a factor of 1.3 when PAR = 2000. Of course, the importance of this term also depends on the magnitude of sqrt(K_sl * X

/ VPD). It is difficult to visualize the differences just from the parameters listed in the text. For the curious audience, it would be helpful to include a supplementary figure comparing optimised and unoptimised LRU_CAP values.

We have added the following figure to the supplement:

[Figure]

*Figure B2. Scatter plots of $LRU_{CAP}$ using the literature values against $LRU_{CAP}$ using the optimized parameter values when assuming (a) infinite or (b) finite mesophyll conductance.*

L296--297: "GPP_{COS,PAR} thus has a stronger radiation response than the other GPP estimates" - It may help to remind the audience that this is caused by lower empirical LRU estimates than the CAP estimates at high light, citing Figure B6.

We have corrected the sentence as "*$GPP_{COS,PAR}$ thus has a stronger radiation response than the other GPP estimates, due to lower empirical LRU estimate than $LRU_{CAP}$ at high PAR (Fig. B6).*"

L312: "Because GPP_{ANN} is based purely on data, it is highly sensitive to uncertainty in the input data." - It may depend on the model architecture and what tricks are used to suppress overfitting and make the model generalisable. Suffice it to say, it is more sensitive to uncertainty in the input data than are parametric partitioning methods.

Thanks, we agree and changed the sentence in "*Because ANN fitting is purely based on the provided examples, $GPP_{ANN}$ could be more sensitive to the uncertainty of (training) data with respect to the parametric partitioning methods.*"

L317--318: "as a different PAR relation was found in Yang et al. (2018) and it is not known if this relation holds elsewhere" - Since the LRU--PAR relationship is empirical, people can choose the equation that best fits their data, be it exponential, power law, or hyperbolic ($y \sim 1/x$). Therefore, it probably won't be a big problem if the empirical equation used in this study does not fit data from other sites well, so long as LRU shows a typical decreasing trend with PAR.

The reviewer is correct. We suggest to change the sentence in "*This PAR relation is site specific and different compared to the one found by Yang et al., (2018). For this reason it is not known if and how it can be used in other sites, where it is suggested to retrieve it directly from observations. The choice of empirical LRU-PAR relation at any given site is to some extent arbitrary.*"

Figure 2: "traditionally partitioned" sounds vague. I would go with "GPP partitioned using a combined nighttime-daytime method".

Corrected as suggested.

Figure 5: "All medians have been calculated using the same number of data points" - This statement seems unclear at first glance. I would rephrase it as "Medians from different methods have been calculated using the same number of data points for each month."
Corrected as *"Median differences have been calculated using the same number of data points for each method in each month."*

Figure B2: "black line" -> "black solid line"
Corrected as suggested.

**Reviewer #2 Georg Wohlfahrt**

Overview:
I am not one of the previous two reviewers, but was asked to review the revised version by the handling editor since one of the original reviewers apparently declined to re-review the revised version. I have quickly glanced over the reply to the reviews and my impression is that the authors have responded adequately to all reviewer comments. Regarding one comment by reviewer #1 on whether the daytime approach assumes respiration is the same during day and night, the authors are correct. While the daytime approach gets the temperature sensitivity from the nighttime data and the base respiration is based on daytime data only, the resulting relationship is applied both day and night (same for nighttime approach).
My own overall assessment of this manuscript is that this is a significant contribution to the literature adding to our understanding of the uncertainty related to partitioning NEE into its component fluxes and especially the differences between CO2 and COS-based approaches. Overall, I have not much to critique and think that the manuscript can be accepted after a few more minor changes listed below. The text is mostly well written, apart from some random switches in tenses, which may however be corrected for during copy-editing.
We thank the reviewer for the comments and suggestions to improve the manuscript.

Detailed comments:
l. 6: also the CO2 flux partitioning is applied to the same boreal forest site – suggest to introduce the site before explaining the 4 approaches
Corrected as suggested: *"In this study, we evaluate four different methods for estimating photosynthesis at a boreal forest at the ecosystem scale, of which two are based on carbon dioxide ($CO_2$) flux measurements and two on carbonyl sulfide (COS) flux measurements."*

l. 14: since the true GPP is unknown this would need to be acknowledge in this sentence, e.g. "… underestimation of midday GPP relative to xy."
Added *"...relative to other GPP methods."*

l. 36: for clarity I would write "… (as in nighttime method) …"
Corrected as suggested.

l. 38: the Lasslop et al. approach is typically referred to as the daytime method/approach for flux partitioning
Corrected as suggested.

l. 42: this was first quantitatively assessed by Wohlfahrt et al. (2005; 10.1016/j.agrformet.2005.02.001)

Reference added.

l. 116-121: this approach amounts to the nighttime flux partitioning logic
Added clarification "*...where R was estimated as in the nighttime method...*"

l. 120: arithmetic mean of some weighting involved?
Corrected as "*arithmetic mean between*"

l. 125: "forces GPP to zero when Ta < 0°C" – the decrease of f(Ta) already starts at positive temperatures (e.g. 5°C) and at Ta = -2 Eq. 4 yields a value of 0.5 – either Eq. 4 is wrong or the sentence in l. 125 should be reformulate to precisely convey how Eq. 4 responds to temperature
We have clarified the sentence as "*...f(T$_a$) is an instantaneous temperature response that brings GPP gradually towards zero at freezing temperatures...*"

l. 129-130: that means you fit the parameters of Eq. 3 to the GPP that was derived from the nighttime flux partitioning (Eq. 1)? If so this should be clearly stated
Yes exactly, this is now clarified as "*The parameters of Eq. 3 were estimated from GPP partitioned with the nighttime method in Eq. 1.*"

l. 156-159: I would start here saying what the average flux was, followed by a description of the (seasonal/diurnal) variability and then continue by saying that the average soil uptake was deducted from the ecosystem-scale flux to yield the canopy net uptake
We have altered the text as suggested: "*Based on previous soil chamber measurements at Hyytiälä forest it is known that the soil COS flux was -2.7 pmol m$^{-2}$ s$^{-1}$ on average with a variation of only 1 pmol m$^{-2}$ s$^{-1}$ during the growing season and a negligible diurnal variation (Kooijmans et al., 2019; Sun et al., 2018a). The average soil flux was thus first subtracted from the quality filtered and gap-filled COS EC fluxes in order to derive the vegetation contribution to the ecosystem COS exchange.*"

l. 159: shouldn't this be "… from the canopy COS fluxes …"?
Yes, corrected as suggested.

l. 162-163: in my view the "respectively" should be placed after the concentrations, not at the end of the sentence
Corrected as suggested.

l. 226: "… may take on negative values due to …"
Corrected as suggested.

l. 228: specify how small
We added that the relative difference was 2% on average during summer months.

l. 254: also boundary layer conductance is assumed infinite in Eq. 8
Finite boundary layer conductance is now also listed as a possible improvement in the model: "*This mismatch suggests there may be scope for further model improvement, such as the inclusion of dark respiration and/or finite mesophyll and boundary layer conductances in the LRU$_{CAP}$ model.*"

l. 286-287: I am not convinced that this will be visible in a plot of NEE as a function of PAR as on the canopy-scale this will be a gradual response as vertically in the canopy leaves/needles experience different PAR levels and thus for the same above-canopy PAR the part of the needles experiencing enough radiation may be inhibited, while others that are more shaded not – this has

been demonstrated with a multi-layer canopy model in Wohlfahrt et al. (2005; 10.1016/j.agrformet.2005.02.001)

The reviewer is correct, and we have added a sentence about this to the revised MS: *"While it is possible that less radiated needles experience less inhibition than well radiated, that cancel out at the ecosystem scale (Wohlfahrt et al., 2005), this test provides some insight to the problem."*

l. 319: in addition there is a scaling issue here – LRU depending in a non-linear fashion on PAR means that vertically in the plant canopy where needles experience PAR that may be different (lower) compared to the above-canopy measurements and thus vertically needles operate at different LRUs; the discussion in l. 339-346 misses this crucial point

We have added a mention about the scaling issue in line 319: *"...it does not take into account the different light conditions inside the canopy, stomatal regulation during drought, or the effects of non-stomatal limitations on photosynthesis."* and a sentence about the possible different LRUs at the end of the section: *"However, it is also possible that LRU varies throughout the canopy due to different light conditions."*

l. 326: "… soil COS exchange …"

Corrected as suggested.

section 4 – conclusions: this section is mostly (l. 348-361) a summary rather than providing conclusions and would profit from expanding the outward looking part (what do these results mean for carbon cycle science?) at the expense of the repetition of the major results (which could be further condensed)

We think that a short summary of the results are still useful in the conclusions, but we shortened the summary and added a new paragraph on the meaning and future steps.

*"Daily $GPP_{ANN}$ and $GPP_{NLR}$ did not differ significantly, and differences were also small on sub-daily and seasonal time scales. $GPP_{COS,PAR}$ was higher than $GPP_{NLR}$ on all time scales studied, including the estimate of three-month cumulative GPP during the peak growing season. In contrast, $GPP_{COS,CAP}$, a new method based on stomatal optimization theory, gave better agreement with $GPP_{NLR}$ on all time scales, and was also less scattered than $GPP_{COS,PAR}$ on a 30-min time scale.*

*The $LRU_{CAP}$ function provides a new theoretical underpinning for COS-based GPP estimates that can be used at other measurement sites, potentially without requiring additional branch chamber measurements. $LRU_{CAP}$ represents a significant improvement on previous LRU functions based on site-specific empirical regressions. However, $LRU_{CAP}$ overestimated LRU at high radiation, when compared to LRU observations at the top of the canopy, leading to a lower midday $GPP_{COS,CAP}$, especially in summer. This discrepancy may result from the assumption of infinite mesophyll conductance, or the absence of dark respiration, in the underlying stomatal optimization model. $LRU_{CAP}$ would benefit from further testing at other measurement sites with COS and $CO_2$ branch flux measurements, including measurements inside the canopy for better canopy-integrated LRU estimates.*

*Although COS flux measurements are noisier, more expensive and more difficult than those of $CO_2$, they provide an opportunity for better process-based understanding of photosynthesis, in comparison with more traditional $CO_2$-based estimates of GPP. In addition to COS, other proxies such as solar induced fluorescence and isotopic flux measurements should be tested simultaneously to properly investigate their deficiencies and advantages in estimating GPP and processes underlying photosynthesis.*

*The establishment of large long-term ecosystem Research Infrastructures (e.g. ICOS, NEON, TERN, see Papale 2020) – involving sites equipped with eddy covariance systems that could potentially also host COS, SIF and isotope sensors – together with the planned launch of the FLEX satellite in 2025 (https://earth.esa.int/eogateway/missions/flex) that will provide global vegetation*

*fluorescence measurements, open up a new phase in monitoring and understanding plant photosynthesis. Our results also underline the important role of small-scale ecophysiological measurements and models in underpinning these larger-scale initiatives."*